# Unleashing Hour-Scale Video Training for Long Video-Language Understanding

Jingyang Lin[1,2*]  Jialian Wu[1] ✉  Ximeng Sun[1]  Ze Wang[1]  Jiang Liu[1]  Yusheng Su[1]
Xiaodong Yu[1]  Hao Chen[1]  Jiebo Luo[2]  Zicheng Liu[1]  Emad Barsoum[1]

[1] AMD   [2] University of Rochester

https://videomarathon.github.io/

## Abstract

Recent long-form video-language understanding benchmarks have driven progress in video large multimodal models (Video-LMMs). However, the scarcity of well-annotated long videos has left the training of hour-long Video-LMMs underexplored. To close this gap, we present **VideoMarathon**, a large-scale hour-long video instruction-following dataset. This dataset includes around 9,700 hours of long videos sourced from diverse domains, ranging from 3 to 60 minutes per video. Specifically, it contains 3.3M high-quality QA pairs, spanning six fundamental topics: temporality, spatiality, object, action, scene, and event. Compared to existing video instruction datasets, VideoMarathon significantly extends training video durations up to 1 hour, and supports 22 diverse tasks requiring both short- and long-term video comprehension. Building on VideoMarathon, we propose **Hour-LLaVA**, a powerful and efficient Video-LMM for hour-scale video-language modeling. It enables hour-long video training and inference at 1-FPS sampling by leveraging a memory augmentation module, which adaptively integrates question-relevant and spatiotemporally informative semantics from the cached full video context. In our experiments, Hour-LLaVA achieves the best performance on multiple representative long video-language benchmarks, demonstrating the high quality of the VideoMarathon dataset and the superiority of the Hour-LLaVA model.

## 1  Introduction

Leveraging strong foundation models [44, 67] and high-quality video instruction-following data [7, 26, 70], recent Video-LMMs [7, 28, 51, 70] have achieved promising performance on basic video-language tasks, such as video question-answering (QA) [60, 61, 66] and video summarization [20, 22, 31, 39, 48]. As attention shifts toward longer video modeling, these models face new challenges in capturing long-term dependencies. To measure progress in this direction, several benchmarks have been introduced for hour-scale video understanding, where the goal is to assess model performance on tasks involving long-form video content over one hour [5, 14, 55, 58]. Early attempts on long-form video-language modeling [45, 46, 50, 68, 74] have shown promising results on these benchmarks.

However, a significant gap remains between the length of videos used during training and those encountered at inference time. While testing videos often exceed one hour [5, 14, 55], most existing training datasets [7, 28, 70] consist of videos shorter than ten minutes, as summarized in Table 1. This mismatch limits the models' ability to explicitly learn long-term dependencies, highlighting the necessity for long-form video instruction-following datasets to bridge this gap.

To this end, we introduce VideoMarathon, a large-scale video instruction-following dataset specifically designed for long-form video-language modeling. VideoMarathon contains around 9,700 hours of

---

*Work was done during the internship at AMD. ✉ Corresponding author: Jialian.Wu@amd.com.

39th Conference on Neural Information Processing Systems (NeurIPS 2025).

Table 1: **Comparison between VideoMarathon and other existing video instruction-following datasets**. OE and MC denote open-ended and multiple-choice, respectively.

| Dataset | Captioner | Summarizer | Total Video Time | Average Video Length | Duration Range | #OE QA | #MC QA |
|---|---|---|---|---|---|---|---|
| LLaVA-Hound [69] | GPT-4V | GPT-4 | 3K hrs | 0.2 mins | 0-8 mins | 900K | 0 |
| ShareGPT4Video [7] | GPT-4V | GPT-4 | 0.2K hrs | 0.3 mins | < 2 mins | 0 | 0 |
| LLaVA-Video-178K [70] | GPT-4o | GPT-4o | 2K hrs | 0.6 mins | < 3mins | 960K | 196K |
| **VideoMarathon (*ours*)** | Qwen2VL-7B | DeepSeek-V3 | **9.7K hrs** | **20.9 mins** | **3-60 mins** | **1.73M** | **1.57M** |

long videos (3-60 min per video) sourced from diverse domains, such as activities [4], egocentric [17], movie [47], cooking [72], and open-world [8] scenarios. Inspired by the recent advances [5, 14, 28, 43], we carefully establish a comprehensive task taxonomy covering six essential topics–temporality, spatiality, object, action, scene, and event–across 22 tasks requiring both short- and long-term comprehension. To generate high-quality video instruction-following QA samples, we develop a hierarchical video captioning pipeline: clip-level descriptions are first produced using Qwen2VL-7B [54] across the six topics, then aggregated into the event- and global-level summaries via DeepSeek-V3 [33]. Based on these hierarchical video captions, we synthesize 3.3M high-quality QA pairs by DeepSeek-V3, guided by task-specific prompts incorporating explicit task definitions and exemplars from prior benchmarks [5, 14, 28, 43]. The resulting QA samples cover both open-ended (OE) and multiple-choice (MC) formats. As shown in Table 1, VideoMarathon substantially surpasses existing video instruction-following datasets in *average video length*, *duration range*, and *scale of QA pairs*.

Due to the limited availability of long video data, most existing Video-LMMs [7, 29, 35, 70] are trained on short videos, typically less than one minute on average (see Table 1). Uniform temporal sampling [27, 70], which selects a fixed number of frames (8-256 per video), works well for short videos but causes substantial information loss when applied to hour-long videos, resulting in severe performance degradation [5, 14, 55]. Moreover, even with large-scale hour-long data available, training existing Video-LMMs on such long videos remains non-trivial, as the quadratic complexity of attention mechanisms [53] causes token budgets to explode. To this end, we propose Hour-LLaVA, a Video-LMM specifically designed for efficient long-form video-language modeling and capable of directly ingesting *hour-long and densely sampled* videos. In particular, Hour-LLaVA stores the full video context sampled at *1 FPS* in a memory repository. To accommodate GPU memory constraints, only a small subset (approximately 6%) of the full video context, referred to as *decayed video tokens*, is used as input to the language model for efficient training and inference. These decayed tokens are then augmented through a memory augmentation (MemAug) module, which adaptively retrieves and integrates spatiotemporally informative and question-relevant semantics from the cached full context. The MemAug module effectively preserves the contextual fidelity of long videos. In summary, Hour-LLaVA achieves a trade-off between *computational efficiency* and *context fidelity*, enabling effective learning from hour-scale video instruction-following data.

Our main contributions are three-fold: (1) we introduce VideoMarathon, the first large-scale hour-long video instruction-following dataset that contains videos ranging from 3 minutes to 1 hour with a total duration of around 9,700 hours and 3.3M diverse QA instructions across 22 tasks; (2) We propose Hour-LLaVA, a Video-LMM capable of processing hour-long videos at 1 FPS in both training and inference, supported by the proposed MemAug mechanism; (3) Training Hour-LLaVA on VideoMarathon, we achieve the best performance among open-source Video-LMMs on four well-known video-language benchmarks, including LVBench [55] (average duration: 4037s), Video-MME [14] (1021s), LongVideoBench [58] (459s), and TempCompass [36] (11s).

## 2 Related Work

**Video Instruction-following Data.** Human annotation is costly and even limits the scale of video instruction-following datasets. Early attempts [25, 40] address the high cost of human annotation by using ASR-generated subtitles, while later methods [18, 27] employ template-based QA generation on existing captioning datasets. However, these approaches often suffer from noisy, low-quality labels. Recent pipelines [7, 69, 70] leverage powerful LMMs (*e.g*., GPT-4V [41]/4o [42]) to produce high-quality video captions and diverse QA pairs. Despite these advancements, most training videos remain short, limiting the models' ability to learn long-term dependencies explicitly. To this end, this work introduces a high-quality, synthetic video instruction-following dataset built on long-form videos, providing a foundation for future research on capturing effective patterns in long videos.

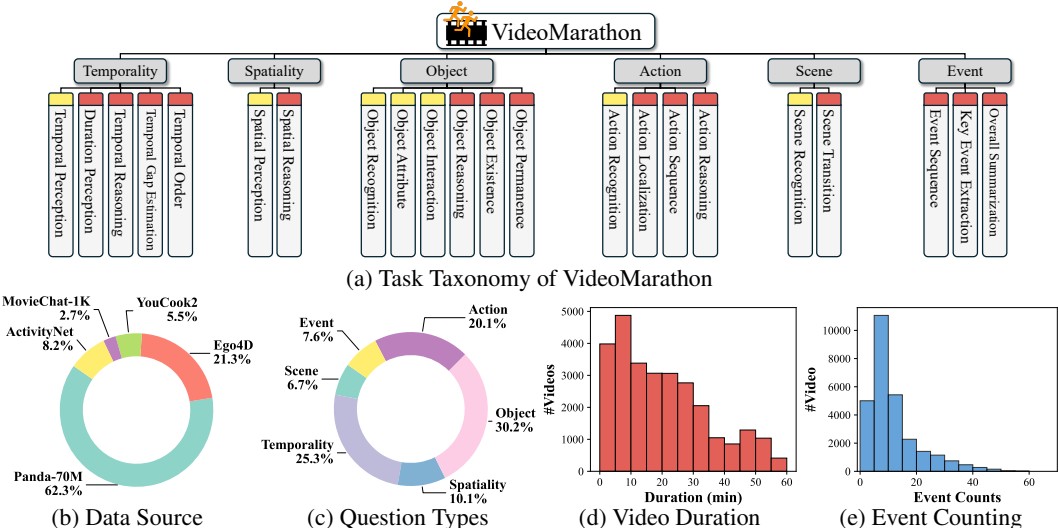

(a) Task Taxonomy of VideoMarathon

(b) Data Source     (c) Question Types     (d) Video Duration     (e) Event Counting

Figure 1: **VideoMarathon: A diverse *long video* instruction-following dataset**. (a) The dataset contains 22 diverse tasks, covering both *short-form* (yellow tag) and *long-form* (red tag) comprehension. (b) The dataset spans diverse video source domains. (c) The dataset features a wide range of question types for long-form video-language modeling. (d) The dataset consists of long videos ranging from three minutes to one hour. (e) The dataset includes complex video content reflected by the number of events per video.

**Long Video-Language Modeling with Language Language Models**. Recent advanced Video-LMMs have shown significant promise in long video-language modeling. Two main approaches facilitate long-form video understanding: Compression-based and extension-based methods. Compression-based methods compress input video tokens across temporal, spatial, and question-guided dimensions, allowing models to process fewer but more informative video tokens. Temporal compression techniques include keyframe selection [45, 50] and slow-fast sampling [13]. Spatial compression involves average pooling [70] and token merging [24]. Other methods [46, 57] apply joint temporal-spatial compression. Question-guided methods [12, 45] retain tokens based on question relevance. Following the success of long-context LLMs [15, 32, 34, 49], extension-based methods expand context windows for long video modeling. LongVA [68] processes over 200K visual tokens, and LongVILA [63] scales to 2M using multimodal sequence parallelism. However, existing approaches confront a fundamental dilemma: compression-based methods prioritize computation affordable and training efficiency, but inevitably suffer from information loss caused by aggressive compression; extension-based methods preserve high-fidelity video context by sequence parallel techniques at the cost of significantly increased training time and GPU usage (*i.e.*, linear growth with video length), particularly for hour-scale videos. To resolve this dilemma, we propose the Hour-LLaVA framework, which intends to prioritize computational efficiency while keeping high-fidelity video context.

## 3 VideoMarathon: A Synthetic Long Video Instruction-following Dataset

This section elaborates on details of constructing VideoMarathon, a long video instruction-following dataset with a total duration of around **9,700 hours**, consisting of **3.3M** QA pairs across **22 tasks**.

**Task Taxonomy**. A comprehensive long video instruction-following dataset requires a well-defined and comprehensive task taxonomy. Inspired by existing video-language benchmarks [5, 14, 28, 43] that cover over 100 tasks, we introduce a comprehensive task taxonomy designed for long video-language understanding. The task taxonomy consists of 22 QA tasks summarized from prior benchmarks and organized into six fundamental topics (see Figure 1a), including temporality, spatiality, object, action, scene, and event. In particular, the temporal topic captures long-term dependencies, while the spatial topic enhances localization. Object and action topics are essential for fundamental video perception, and the scene topic offers contextual grounding and transition cues. Event topic supports long-range understanding of key video content. This task taxonomy encourages Video-LMMs to develop both *long-form* and *short-form* video comprehension. Please refer to Figure 11 for detailed descriptions of the above 22 QA tasks in Appendix A.

**Video Sources**. High-quality and diverse video content is essential for effective video-language modeling. To ensure a sufficient amount of long videos while maintaining quality and diversity, VideoMarathon integrates five representative public video datasets: Panda-70M [8], Ego4D [17], ActivityNet [4], YouCook2 [72], and MovieChat-1K [47]. Figure 1b presents the proportions of these video sources. Panda-70M constitutes the majority of VideoMarathon, covering diverse topics such as daily activities, sports, science, arts, transportation, travel, education, wildlife, entertainment, and industry. Domain-specific datasets further enrich the collection: Ego4D captures real-world activities from a first-person view; ActivityNet covers a broad range of human actions; YouCook2 focuses on cooking scenarios; and MovieChat-1K contributes content from movies and TV series. To enhance temporal complexity within videos, we select videos with at least three distinct events. Figure 1e shows the distribution of event counts per video. In total, VideoMarathon comprises 28K videos across diverse domains, supporting long-form video-language modeling.

**Hierarchical Video Captioning**. In contrast to short-form video QA tasks that typically rely on concise captions, long-form video QA demands hierarchical contextual understanding, including detailed clip-wise content, event-level structures, and global narratives. To meet this need, we introduce a hierarchical video captioning pipeline for capturing both fine-grained and high-level semantics over extended temporal ranges. The resulting captions serve as essential inputs for generating reliable, diverse, and context-rich QA training samples tailored to long-form video-language understanding.

- **Clip-Level Video Captioning**. Instead of generating a single brief summary for each video clip [7, 63, 70], VideoMarathon provides detailed descriptions (see Figure 12a) for each video clip from six perspectives: temporality, spatiality, object, action, scene, and overall summary. These chronologically ordered descriptions serve as reliable contexts for the subsequent diverse QA generation. Technically, we leverage Qwen2VL-7B [54], a powerful yet lightweight LMM, as the clip-level captioner, following the prior practice [74].

- **Event-Level Video Captioning**. We first identify event boundaries (*i.e.*, start and end timestamps) based on the chronologically ordered clip-level captions. Once the event boundaries are determined, all detailed captions within each event are then aggregated into a comprehensive event-level summary. Specifically, we use DeepSeek-V3 [33] to summarize the six-perspective clip-level captions along with adjacent event descriptions into a cohesive event-level summaries. As shown in Figure 12c, this process results in detailed event-level video captions, each associated with its corresponding event boundaries.

- **Global-Level Video Captioning**. This step focuses on generating a natural-toned and global-level description for the entire video. Specifically, we feed the event-level captions along with their boundaries into DeepSeek-V3, obtaining a cohesive and comprehensive video summary. Figure 12d shows that these captions provide global context for the subsequent QA generation.

Appendix A provides the detailed prompts used for the hierarchical video captioning, including Figure 5 for clip-level video captioning, Figure 6 for event splitting, Figure 7 for event-level video captioning, and Figure 8 for global-level video captioning.

**Diverse QA Generation for Long Videos**. Leveraging the multi-level video captions, we generate high-quality and diverse QA pairs across 22 tasks, spanning six major topics in both open-ended and multiple-choice formats. To achieve this, we design topic-specific prompts that incorporate (1) detailed instructions for QA generation, (2) topic-specific task descriptions (*e.g.*, scene recognition and transitions within the scene topic), (3) clip-level captions of the corresponding topic following the chronological order, (4) the comprehensive global-level captions, and (5) high-quality QA demo examples sourced from established video benchmarks [5, 14, 28, 43]. These structured prompts facilitate the creation of contextually grounded and diverse QA pairs that reflect the unique challenges of long-form video understanding. Detailed prompt templates for open-ended (Figure 9) and multiple-choice (Figure 10) QA generation are provided in Appendix A.

**Comparison**. Table 1 presents a comparison between our proposed VideoMarathon dataset and existing video instruction-following datasets [7, 70]. The most significant distinction is VideoMarathon's substantially longer total video time and average video length. Moreover, VideoMarathon covers a broader duration range from 3 to 60 minutes, effectively bridging the gap in *long-form* video instruction data. Also, the diverse QA tasks and the balanced distribution of open-ended and multiple-choice QA pairs enable Video-LMMs to better handle a wide range of challenging real-world questions.

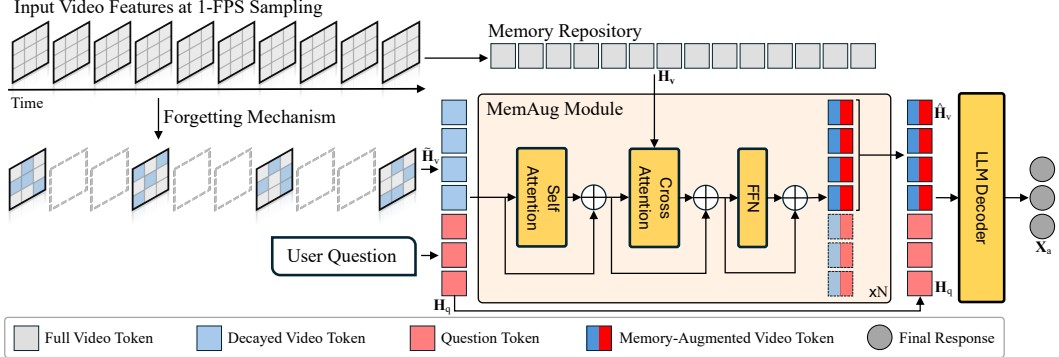

Figure 2: **Overview of the Hour-LLaVA Framework.** Input video features $\mathbf{H}_v$ encoded from 1-FPS sampled frames are selectively decayed spatially and temporally through a forgetting mechanism, producing decayed video tokens $\tilde{\mathbf{H}}_v$ for efficient video modeling. Meanwhile, full video features $\mathbf{H}_v$ are stored in a memory repository. Given the decayed tokens $\tilde{\mathbf{H}}_v$ and a user question tokens $\mathbf{H}_q$, the MemAug module enhances them with full video context and user question-relevant details from the memory repository, obtaining memory-augmented video tokens $\hat{\mathbf{H}}_v$. These augmented tokens are then passed with the original user question tokens $\mathbf{H}_q$ into the LLM decoder to generate the final response $\mathbf{X}_a$.

## 4 Hour-LLaVA: An Efficient Hour-scale Video-LMM

**Problem Formulation**. Standard video QA tasks require video-language models to generate an answer $\mathbf{X}_a$, based on a given input video $\mathbf{X}_v$ and a user question $\mathbf{X}_q$ by modeling the conditional probability of $p_\theta(\mathbf{X}_a|\mathbf{X}_v, \mathbf{X}_q)$, where $\theta$ is the parameter of video-language models. More specifically, the input video is first fed into a vision encoder $f_{\theta_V}(\cdot)$ parameterized by $\theta_V$ to produce the visual feature $\mathbf{Z}_v = f_{\theta_V}(\mathbf{X}_v)$. Then, a projector $\mathbf{W}$ converts the visual feature $\mathbf{Z}_v$ to a visual embedding $\mathbf{H}_v = \mathbf{W} \cdot \mathbf{Z}_v$ in the language embedding space. Meanwhile, the input user question $\mathbf{X}_q$ is projected to language embeddings $\mathbf{H}_q$ by a word embedding model. Finally, the language model $f_{\theta_L}(\cdot)$ predicts the answer $\mathbf{X}_a = f_{\theta_L}(\mathbf{H}_v, \mathbf{H}_q)$, where $\theta_L$ is the parameter of the language model.

**Memory Augmentation for Long Video Modeling**. Modeling hour-long videos poses a challenge, as densely processing all frames with LLMs is impractical under GPU memory constraints. To address this challenge, we draw inspiration from the human memory system, which selectively retains and recalls essential past experiences while discarding irrelevant or redundant information, striking a balance between efficiency and comprehensiveness [2, 11]. Motivated by this principle, we design a memory augmentation mechanism to enable hour-scale video-language understanding. This mechanism consists of three main components: a memory repository, a forgetting mechanism, and a MemAug module. Together, they allow the model to operate on compressed representations while maintaining access to the full video context.

- **Memory Repository**. High-fidelity video features $\mathbf{H}_v$ extracted at 1 FPS are stored in a memory repository serving as long-term memory, enabling the model to retain complete video context without requiring the LLM decoder to process every frame.

- **Forgetting Mechanism**. Due to the GPU memory constraints, we employ a forgetting mechanism $\mathcal{M}_{forget}$ to compresses the full video tokens $\mathbf{H}_v$ into a reduced set of *decayed video tokens* $\tilde{\mathbf{H}}_v = \mathcal{M}_{forget}(\mathbf{H}_v)$ by selectively discarding the tokens in both spatial and temporal dimensions. This prevents the training and inference cost from growing linearly with video length. In practice, the forgetting mechanism $\mathcal{M}_{forget}$ can be implemented by various token compression strategies [23, 45, 46, 70]. Section 5.4 provides a comprehensive comparison among different forgetting mechanisms.

- **MemAug Module**. The MemAug module dynamically recovers the informative video tokens discarded during the forgetting stage from the memory repository. Without the MemAug module, forgetting mechanism alone would inevitably incur irreversible context loss and lead to sub-optimal performance. Technically, the MemAug module is implemented using standard transformer blocks [53]. In cross-attention, the decayed video tokens $\tilde{\mathbf{H}}_v$ and user question tokens $\mathbf{H}_q$ (as queries) attend and retrieve relevant information from the memory repository $\mathbf{H}_v$ (as keys and values). In self-attention, question-relevant video information flows from the user question tokens

$\mathbf{H}_q$ to the decayed video tokens $\tilde{\mathbf{H}}_v$. Through this process, the decayed video tokens $\tilde{\mathbf{H}}_v$ are enriched with both full video context and question-specific details, resulting in *memory-augmented video tokens* $\hat{\mathbf{H}}_v$ that support long-dependence modeling for video understanding:

$$\hat{\mathbf{H}}_v = f_{\theta_M}(\tilde{\mathbf{H}}_v, \mathbf{H}_q | \mathbf{H}_v), \tag{1}$$

where $\theta_M$ denotes the parameters of the MemAug module $f_{\theta_M}(\cdot)$.

**Hour-LLaVA**. Powered by memory augmentation, we propose Hour-LLaVA, an efficient video-language model capable of modeling hour-long videos at 1 FPS. It comprises three key modules: a video encoder, a memory augmentation module (*i.e.*, MemAug), and an LLM decoder. Figure 2 shows the Hour-LLaVA framework, with the video encoder omitted for simplicity.

- **Video Encoding**. Following existing Video-LMMs [70, 74], we adopt SigLIP [67] followed by a vision-language projector for video encoding. Specifically, we sample video frames at 1 FPS and extract visual features $\mathbf{Z}_v$, which are then average-pooled to a fixed spatial resolution of $8 \times 8$ per frame. We then project the average-pooled visual features into the language embedding space via a two-layer MLP with GELU [21] activation, obtaining the full video tokens $\mathbf{H}_v$.

- **MemAug**. The decayed video tokens $\tilde{\mathbf{H}}_v$ are obtained by "forgetting" $\sim$94% of the full video tokens with an overall compression ratio of $\frac{1}{16}$ through the forgetting mechanism, achieved by discarding tokens with a $\frac{1}{4}$ compression ratio along both spatial and temporal dimensions. These decayed tokens are then enriched with the full video tokens via the MemAug module, implemented with $N = 4$ transformer blocks. To associate the spatial-temporal correspondence between the full video tokens in the memory repository and the decayed tokens, we first flatten the full video tokens into 1D embeddings and apply 1D structured RoPE, following prior works [16, 49]. Based on their spatial-temporal alignment, we then construct a corresponding 1D RoPE for the decayed tokens, ensuring that each decayed token (query) carries positional information consistent with its related full tokens (keys and values) during cross-attention.

- **LLM Decoder**. Given the memory-augmented video tokens $\hat{\mathbf{H}}_v$ and the user question tokens $\mathbf{H}_q$, LLM decoder generates final response $\mathbf{X}_a$. We employ Qwen2.5-3B-Instruct [65] and Qwen2-7B-Instruct [64] as the LLM decoders for Hour-LLaVA-3B and Hour-LLaVA-7B, respectively.

**Training Schedules**. The training schedules of Hour-LLaVA follow three stages: image-language pretraining, video-language adaptation, and video instruction tuning. Table 8 presents more details of the training schedules.

- **Image-Language Pretraining**. We perform image-language pretraining with 3B image-text pairs from the single-image subset of LLaVA-OV [26]. Full-resolution image tokens are stored in the memory repository, from which $\frac{1}{4}$ tokens are retained as decayed visual tokens via the spatial forgetting mechanism. Only the Transformer blocks of MemAug are trained for one epoch.

- **Video-Language Adaptation**. Following the data composition that adheres to established practices [70, 74], the model is adapted to video-language inputs using a small amount of mixture of 0.12M single-image, 0.05M multi-image, and 0.09M text data from LLaVA-OV, plus 0.3M short video samples from LLaVA-Video-178K [70]. A $\frac{1}{4}$ token compression is applied in both spatial and temporal dimensions via the forgetting mechanism. All model parameters are trained for one epoch. This stage intends to slightly adapt all model parameters to the video domain, thereby constructing a strong baseline for the subsequent long-video training stage.

- **Video Instruction Tuning**. This stage uses instruction-following data involving long video content. For training efficiency, up to five QA pairs from the same VideoMarathon video are grouped as a multi-turn conversation. The corpus includes 1.14M single-image, 0.5M multi-image, and 0.81M text samples from LLaVA-OV, 1.3M short video samples from LLaVA-Video-178K, and 0.7M long video samples from VideoMarathon. The same forgetting mechanism is applied as the previous stage. The vision encoder is frozen, while the remaining modules are fine-tuned.

## 5 Experiments

### 5.1 Experimental Setting

**Evaluation Benchmarks**. We evaluate our models on four mainstream video-language benchmarks: TempCompass [36], LongVideoBench [58], Video-MME [14], and LVBench [55]. **TempCompass** focuses on assessing the temporal reasoning ability of Video-LMMs for short videos. **LongVideoBench**

Table 2: **Performance comparison of existing LMMs** on TempCompass, LongVideoBench, VideoMME, and LVBench datasets. M-Avg denotes the average performance of multiple-choice tasks. [Blue] boxes denote the average durations of benchmarks. [Red] highlights that LVBench's average video length exceeds the maximum length in the training stage. The symbol † marks our reimplemented results. **Bold** font denotes the best performance among models at the same scale, while underline indicates the second-best.

| Method | LLM Params | Input Video | TempCompass M-Avg [11s] | LongVideoBench M-Avg [459s] | VideoMME (w/o & w/ subtitles) Overall [1021s] | Medium [516s] | Long [2466s] | LVBench Avg [4037s] |
|---|---|---|---|---|---|---|---|---|
| **Proprietary LMM** | | | | | | | | |
| GPT-4V [41] | - | 10 frames | - | 61.3 | 59.9/63.3 | 55.8/59.7 | 53.5/56.9 | - |
| GPT-4o [42] | - | 384 frames | 70.9 | 66.7 | 71.9/77.2 | 70.3/76.6 | 65.3/72.1 | 48.9 |
| Gemini 1.5 Flash [52] | - | 0.5/1 fps | - | 61.6 | 70.3/75.0 | 68.8/74.7 | 61.1/68.8 | - |
| Gemini 1.5 Pro [52] | - | 0.5/1 fps | 69.3 | 64.0 | 75.0/81.3 | 74.3/81.0 | 67.4/77.4 | 33.1 |
| **Open-source LMM (<7B)** | | | | | | | | |
| VILA1.5-3B [30] | 3B | 8 frames | 56.1 | 42.9 | 42.2/44.2 | - | - | - |
| Phi-3.5-Vision-4.2B [1] | 4.2B | 16 frames | - | - | 50.8/ - | - | - | - |
| LongVU-3.2B [45] | 3.2B | 1 fps | - | - | - /51.5 | - | - /47.2 | - |
| InternVL2.5-2B [9] | 2B | 64 frames | 53.4 | 46.0 | 51.9/54.1 | - | - | - |
| Apollo-1.5B [74] | 1.5B | 2 fps | 60.8 | 54.1 | 53.0/54.6 | - | - | - |
| Apollo-3B [74] | 3B | 2 fps | 62.5 | 55.1 | 58.4/60.6 | - | - | - |
| LLaVA-OV-SI-3B† [26] | 3B | 32 frames | 55.6 | 49.6 | 51.1/54.5 | 49.2/52.1 | 44.1/46.4 | 35.4 |
| LLaVA-Video-3B† [70] | 3B | 64 frames | 63.4 | 55.2 | 58.7/60.7 | 55.2/57.3 | 47.0/49.9 | 41.7 |
| *Hour-LLaVA-3B (ours)* | 3B | 1 fps | **63.6** | **57.8** | **60.6/66.7** | **59.0/65.4** | **52.1/60.4** | **44.7** |
| **Open-source LMM (7-8B)** | | | | | | | | |
| Video-LLaVA [29] | 7B | 8 frames | 37.9 | 39.1 | 39.9/41.6 | 38.0/40.7 | 36.2/38.1 | - |
| VideoChat2 [28] | 7B | 16 frames | 51.1 | 39.3 | 39.5/43.8 | 37.0/39.4 | 33.2/39.2 | - |
| ShareGPT4Video [7] | 8B | 16 frames | 59.4 | 41.8 | 39.9/43.6 | 36.3/39.3 | 35.0/37.9 | - |
| VideoLLaMA2 [10] | 7B | 16 frames | - | 51.4 | 47.9/50.3 | 37.0/39.4 | 33.2/39.2 | - |
| Video-XL [46] | 7B | 1 fps | - | 50.7 | 55.5/61.0 | - | - | - |
| Kangaroo [35] | 8B | 64 frames | 62.5 | 54.8 | 56.0/57.6 | 55.3/55.4 | 46.7/49.3 | 39.4 |
| LongVA [68] | 7B | 128 frames | - | - | 52.6/54.3 | 50.4/53.6 | 46.2/47.6 | - |
| LongVILA [63] | 7B | 256 frames | - | - | 60.1/65.1 | 58.3/64.9 | 53.0/57.4 | - |
| LongVU [45] | 7B | 1 fps | - | /60.9 | - | - | - /59.5 | - |
| Apollo-7B [74] | 7B | 2 fps | 64.9 | 58.5 | 61.3/63.3 | - | - | - |
| LLaVA-OV-SI-7B† [26] | 7B | 32 frames | 57.2† | 52.1† | 58.2/61.5 | 52.6†/55.4† | 47.9†/50.2† | 36.0† |
| LLaVA-Video-7B† [70] | 7B | 64 frames | 64.3† | 58.2 | 63.3/69.7 | 58.9/62.9† | 53.0/55.0† | 42.2† |
| *Hour-LLaVA-7B (ours)* | 7B | 1 fps | **68.1** | **60.4** | **63.6/70.2** | **63.8/70.0** | **55.0/65.1** | **45.6** |

consists of varying-length web-collected videos up to one hour with their subtitles across diverse themes, evaluating the abilities in retrieving and reasoning over detailed information from long videos. **Video-MME** is a comprehensive multimodal benchmark designed to evaluate long video understanding across diverse video types and temporal spans. **LVBench** challenges models to demonstrate long-term memory and extended comprehension across multimodal inputs.

**Implementation Details**. Following existing practices [45, 68, 70], we initialize our Hour-LLaVA models with pretrained Image-LMMs. Specifically, the vision encoder and the LLM decoder of Hour-LLaVA-7B are initialized from LLaVA-OV-SI-7B [26], which is trained solely on image data. Due to the absence of LLaVA-OV-SI-3B, we pretrain a 3B version of LLaVA-OV-SI using Qwen2.5-3B-Instruct model, and then initialize Hour-LLaVA-3B from it. For video-language training, we set the global batch sizes to 128 and 256 for the 3B and 7B models, respectively. A learning rate of 2e-5 is used with a 0.03 warmup ratio under a cosine annealing schedule. The models are optimized using the AdamW [37] optimizer with a cross-entropy loss. We train Hour-LLaVA-3B with 64 AMD MI250 GPUs and Hour-LLaVA-7B with 64 AMD MI300X GPUs, respectively. We conduct all the ablation studies using Hour-LLaVA-3B in Section 5.3 and 5.4. Please refer to Appendix A.3 for more implementation details.

## 5.2 Main Results

**Overview**. As shown in Table 2, Hour-LLaVA consistently achieves the best performance on four well-established video benchmarks in both the 3B and 7-8B model size categories. Notably, Hour-LLaVA-3B even surpasses more than half of the current 7-8B Video-LMMs.

**TempCompass**. The results on TempCompass show that Hour-LLaVA maintains strong performance on short-form video-language tasks, even after introducing long video-language training samples.

**LongVideoBench**. Hour-LLaVA demonstrates state-of-the-art performance among open-source models across both the 3B and 7B parameter scales on LongVideoBench. Specifically, Hour-LLaVA-3B and Hour-LLaVA-7B outperform the second-best model by 2.6 and 1.9 points, respectively.

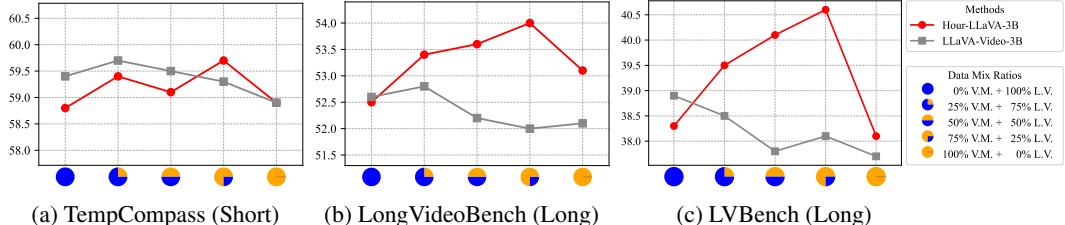

(a) TempCompass (Short)  (b) LongVideoBench (Long)  (c) LVBench (Long)

Figure 3: **Dataset ablation and methodology comparison**. The analysis is evaluated across three benchmarks: (a) TempCompass, (b) LongVideoBench, and (c) LVBench. It presents the performance of Hour-LLaVA-3B and LLaVA-Video-3B models. The x-axis represents different training data mixture configurations, with exact ratios indicated in the legend. V.M. and L.V. refer to the VideoMarathon and LLaVA-Video-178K datasets.

**VideoMME**. Hour-LLaVA consistently outperforms other open-source models at both the 3B and 7B scales on VideoMME. Remarkably, Hour-LLaVA achieves significantly higher performance in both medium- and long-length video settings, highlighting its strong capability in long video understanding

**LVBench**. The maximum video length of LVBench is more than two hours, and its average video length (67 minutes) already exceeds the maximum video length in our training (60 minutes). As such, this benchmark serves as a pressure test for long video understanding and highlights the model's ability to generalize beyond its training distribution. As shown in Table 2, Hour-LLaVA achieves leading performance, outperforming the second-best models by 3.0 and 3.4 points under the 3B and 7B settings, respectively.

**Comparison with Baseline Models**. LLaVA-OV-SI [26] serves as the image-language pretrained model for Hour-LLaVA, while both LLaVA-Video [70] and Hour-LLaVA adopt the same LLM decoder. The results show that Hour-LLaVA considerably enhances the performance of its pretrained model across both 3B and 7B model scales. Furthermore, Hour-LLaVA consistently outperforms its strong peer, LLaVA-Video, especially in medium- and long-video scenarios.

### 5.3 Analysis on VideoMarathon Dataset

**Comparison between VideoMarathon and LLaVA-Video-178K.** We compare our proposed Video-Marathon dataset with LLaVA-Video-178K [70], the largest publicly available video instruction-tuning dataset to date. For a fair comparison, we construct two subsets from VideoMarathon (V.M.) and LLaVA-Video-178K (L.V.), matched in both the number of videos and the number of video-language instruction samples. Specifically, we randomly select 70K instruction-following samples over 10K videos from each dataset. To support multimodal learning, both datasets are further combined with the same 30K single-image, 10K multi-image, and 20K text-only samples from LLaVA-OV [26].

Figure 3 illustrates how performance varies with different mixtures of V.M. (long-video) and L.V. (short-video) training data. On the far left of the x-axis, the model is trained solely on 70K short-video samples from L.V.; moving right, short-video samples are gradually replaced by long-video samples from V.M., reaching 100% V.M. on the far right. The pie charts visualize the data mix ratios for each setting. As shown in Figures 3b and 3c, Hour-LLaVA (—•—) achieves progressively better performance as the proportion of long-video data increases, peaking when the long-to-short ratio reaches 3:1, and then declining once L.V. data is entirely absent. These results highlight the crucial role of VideoMarathon in enhancing long video-language understanding. They also suggest that mixing different datasets can be beneficial even when the training and testing video lengths differ, possibly due to the increased diversity introduced by heterogeneous data sources. In addition, Figure 3a shows that incorporating long videos from VideoMarathon in training does not considerably affect performance on short video-language understanding. In the following ablation experiments, we use the same data recipe, along with a mixture of long- and short-video at a 3:1 ratio.

**Sparse Sampling Limits Long Video-Language Learning.** Using the VideoMarathon dataset, we also train LLaVA-Video [70], a representative Video-LMM that adopts sparse temporal sampling by uniformly selecting 64-frame video features as input to the LLM decoder. As shown in Figure 3, LLaVA-Video (—■—) fails to benefit from an increased proportion of long-video samples, with its performance even declining after long-video training. This result indicates that *sparse sampling*

Table 3: Comparison of spatial forgetting strategies.

| Spatial Forgetting | Tokens/img | MMStar | Sci.QA | R.W.QA |
|---|---|---|---|---|
| LLaVA-OV-SI-3B (Base) | 729 | **52.8** | **84.7** | 58.8 |
| + Random | 196 | 51.9 | 84.5 | **59.6** |
| + Uniform | 196 | 51.5 | 83.7 | 59.5 |

Table 4: Comparison of temporal forgetting strategies.

| Temporal Forgetting | LongVideoBench | LVBench |
|---|---|---|
| Uniform | **54.0** | **40.6** |
| Keyframe | 53.5 | 40.3 |
| Question-guided | 53.4 | 39.7 |
| Random | 53.2 | 38.8 |

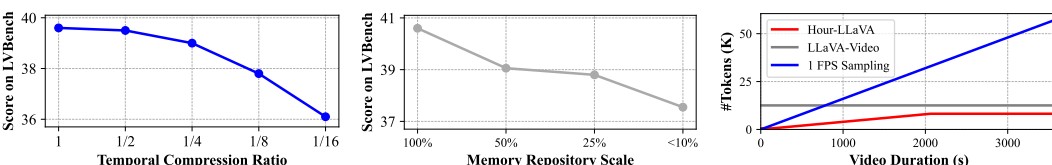

Figure 4: Impact of compression ratio for temporal forgetting (*left*), and memory repository scale (*middle*). Comparison of the number of visual tokens input to the LLM decoder (*right*).

*is insufficient to capture the effective patterns required for long video-language understanding*. In contrast, Hour-LLaVA (—•—) effectively achieves long-video language modeling by leveraging access to a memory repository that stores full dense video content at 1 FPS. These results highlight the urgent demands for new training paradigms tailored to long video–language modeling, and position VideoMarathon as a practical platform for developing them.

## 5.4 Analysis on Key Components of Hour-LLaVA

**Forgetting Mechanisms**. The forgetting mechanism is proposed to compress full video tokens into decayed video tokens to reduce the number of tokens processed by the LLM decoder. The decayed tokens will then be augmented by the MemAug module to retain or recall the informative semantics. The forgetting mechanism can be implemented with different token compression strategies [23, 45, 46, 70] in both spatial and temporal domains. We ablate different forgetting mechanisms guided by the MemAug module as follows:

- **Spatial Forgetting** (SF). We compare two straightforward methods: uniform SF (discarding tokens at regular intervals over the 2D image) and random SF (discarding tokens randomly) with a compression ratio of $1/4$. We conduct this ablation during the image-language pretraining stage and evaluate on three popular image-language benchmarks: MMStar [6], ScienceQA [38] (Sci.QA), and RealWorldQA [59] (R.W.QA). Notably, despite using only $1/4$ tokens, both SF strategies achieve comparable performance to the base model without token compression as shown in Table 3. It highlights the success of our MemAug module for token compression in the image domain, suggesting that MemAug can also be applied to Image-LMMs for token compression. For Hour-LLaVA, we adopt random SF with $1/4$ compression ratio for its simplicity and effectiveness.

- **Temporal Forgetting** (TF). We evaluate four temporal compression approaches: random, uniform [23, 35, 62, 70], keyframe-based [45, 50], and question-guided temporal compression [45, 46]. All strategies are evaluated using a compression ratio of $1/4$. As shown in Table 4, uniform TF yields the best overall performance, suggesting that the MemAug module is effective without relying on manually designed compression strategies. Furthermore, we analyze the effect of different compression ratios for temporal forgetting on LVBench, ranging from 1 down to $1/16$ as shown in Figure 4 (*left*). We adopt the uniform TF with a ratio of $1/4$ for Hour-LLaVA, balancing the trade-off between performance and efficiency introduced by temporal compression.

- **Token Control for Videos with Extreme Lengths**. Although forgetting mechanisms significantly reduce the tokens processed by the LLM decoder, they remain insufficient for extremely long videos (*e.g.*, 24-hour videos). We set the maximum number of retained frames to 512 so as to constrain the computational cost. Conversely, for extremely short videos, we enforce a minimum of 32 retained frames to preserve sufficient contextual information. Note that these thresholds apply to decayed tokens only, which means Hour-LLaVA still accesses the full video context at 1-FPS sampling via memory repository. Figure 4 (*right*) compares the number of visual tokens fed into the LLM decoder across Hour-LLaVA, LLaVA-Video, and vanilla 1-FPS sampling methods. It highlights that Hour-LLaVA consistently uses fewer tokens than others.

Table 5: Impact of MemAug module on different token compression strategies.

| Token Compression | MemAug | LongVideoBench | LVBench |
|---|---|---|---|
| Uniform | ✓ | 54.0 (+1.9) | 40.6 (+2.3) |
| | × | 52.1 | 38.3 |
| Keyframe | ✓ | 53.5 (+1.5) | 40.3 (+1.4) |
| | × | 52.0 | 38.9 |
| Question-guided | ✓ | 53.4 (+2.4) | 39.7 (+2.2) |
| | × | 51.0 | 37.5 |

Table 6: Impact of the number of MemAug blocks. M.A. refers to MemAug.

| # M.A. Blocks | TempCompass | LongVideoBench | LVBench |
|---|---|---|---|
| 1 | 59.4 | 53.2 | 40.0 |
| 2 | **59.8** | 53.5 | 40.3 |
| 4 | 59.7 | **54.0** | **40.6** |
| 8 | 59.2 | 53.4 | 40.5 |

**MemAug Module**. The MemAug module is designed to dynamically recover the informative video tokens from the memory repository. To retain informative tokens from full videos, previous studies have explored various video token compression approaches [12, 45, 50]. However, these approaches typically rely on hand-crafted rules (*e.g.*, predefined thresholds) and inevitably suffer from irreversible information loss. In contrast, the MemAug module performs video context compression through a *learnable process* supervised by video instruction data.

- **Impact of MemAug under Different Compression Strategies**. To validate the effectiveness of MemAug, we apply MemAug module on several representative video token compression methods [12, 45, 50], including uniform [23, 35, 62, 70], keyframe-based [45, 50], and question-guided [45, 46] temporal compression. For fairness, all methods feed the same total number of compressed tokens to the LLM decoder. Specifically, we apply random spatial compression with a ratio of $^1/_4$ and various temporal compression techniques with a ratio of $^1/_4$ on 1-FPS input videos. As shown in Table 5, models equipped with MemAug consistently achieve 1.4-2.4% empirical gains over those without it across all token compression strategies, demonstrating that *MemAug effectively re-injects informative cues*. Without MemAug, conventional token compression approaches incur irreversible context loss, resulting in sub-optimal performance. As a result, by coupling MemAug with forgetting mechanism, Hour-LLaVA delivers compression-level efficiency while still preserving the high-fidelity video content.

- **Robustness of MemAug Module**. Notably, Table 5 presents that the performance with MemAug remains comparable regardless of the underlying token compression approaches, indicating that *MemAug is robust and largely insensitive to the choice of token forgetting mechanism*. This consistency demonstrates that MemAug effectively adapts to distinct temporal compression paradigms, ensuring reliable information recovery even under different token compression settings.

- **Effect of the number of MemAug blocks**. We evaluate 1, 2, 4, and 8 MemAug blocks on three benchmarks. Table 6 demonstrates that performance improves up to 2 or 4 blocks and then decreases slightly at 8. More specifically, for short videos (TempCompass), the sweet spot lies in the range from 2 to 4 blocks, while for medium- and long-form videos (LongVideoBench and LVBench), 4 blocks provides the optimal results. Consequently, we adopt *4 MemAug blocks* as the default.

**Memory Repository**. We investigate the impact of the memory repository scale on performance on LVBench. Figure 4 (*middle*) shows that the performance of Hour-LLaVA decreases as the memory repository scale becomes smaller. This trend indicates that *information loss in the memory repository limits the capacity of long video understanding*. Therefore, to preserve as much temporal information as possible, we retain video tokens sampled at 1 FPS in the memory repository.

Please refer to the Appendix A.3.2 for more details of the above experiments.

## 6 Conclusion

In this study, we introduce VideoMarathon, a large-scale video instruction-following dataset. Comprising long videos with a total duration of around 9,700 hours and 3.3M QA pairs, VideoMarathon covers 22 challenging tasks that require both short- and long-term video understanding. Building on VideoMarathon, we propose Hour-LLaVA, an efficient and powerful Video-LMM optimized for hour-scale video modeling. By leveraging a memory augmentation (MemAug) mechanism, Hour-LLaVA effectively integrates information from the full video context while maintaining efficiency through 1-FPS sampling. Extensive evaluations on several video-language benchmarks validate the high quality of the VideoMarathon dataset and the superiority of the Hour-LLaVA model.

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

# A Appendix

This document provides more implementation details of the VideoMarathon dataset construction, additional experimental details, limitations, and broader impacts, organized as follows:

- **Details of VideoMarathon Construction** (Section A.1). We provide the exact prompts and examples for hierarchical video captioning and topic-specific QA generation.
- **Human Evaluation on VideoMarathon** (Section A.2). We conduct the human evaluation on the proposed VideoMarathon dataset.
- **More Experimental Details** (Section A.3). We present additional experimental details, including detailed training schedules for Hour-LLaVA and experimental settings of ablation studies in Sections 5.3 and 5.4.
- **Additional Empirical Analysis** (Section A.4). We provide additional empirical analysis to support the superiority of the proposed Hour-LLaVA framework.
- **Limitations** (Section A.5). We discuss several limitations of this study.
- **Broader Impacts**(Section A.6). We analyze both the potential positive societal impacts and the negative societal impacts of this work.

## A.1 Details of VideoMarathon Construction

### A.1.1 Details of Hierarchical Video Captioning

**Prompts.** We provide the exact prompts used for hierarchical video captioning, including clip-level captioning (Figure 5), event-level captioning (Figure 7), and global-level captioning (Figure 8). Additionally, the prompt used for event splitting is presented in Figure 6.

**Examples**. Figure 12 presents several examples of hierarchical video captions, including clip-level, event-level, global-level video captions, and event splits. In addition, Figure 13 presents the word cloud of all the global-level video descriptions in the VideoMarathon dataset.

### A.1.2 Details of QA Generation

**Prompts**. We present the exact prompts used for topic-specific QA generation, including the open-ended (OE) QA prompt in Figure 9 and the multiple-choice (MC) QA prompt in Figure 10. Also, the detailed descriptions of the 22 sub-tasks across six core topics are provided in Figure 11. Please refer to the data file for exact QA examples from the VideoMarathon dataset.

## A.2 Human Evaluation on VideoMarathon

To quantify the data quality, we manually check 330 randomly selected VideoMarathon QA pairs, including 80 multiple-choice and 250 open-ended samples.

For the **Multiple-Choice (MC) QA** samples, we assess the accuracy of the synthetic samples by assigning a score of 1 if there is exactly one correct answer. A score of 0 is given if the question is irrelevant to the video, if no correct option exists, or if multiple options are correct. The human evaluation on 80 samples achieves an accuracy of 78.7%, which demonstrates the reliability of the synthetic multiple-choice QA samples.

For the **Open-Ended (OE) QA** samples, we validate the data quality from two perspectives:

- **Yes/No Bias**: Yes/No Bias evaluation [19] validates the extent to which the models tend to respond with "yes". In particular, the evaluation consists of "Yes Percentage Difference" (Pct. Diff) and "False Positive Ratio" (FP Ratio) metrics. Smaller Pct. Diff (closer to 0.0) reveals less bias, which means that the predicted number of "Yes" is closer to the ground truth. FP Ratio measures how likely the model responds with "Yes" out of all incorrect responses, where a value closer to 0.5 indicates less bias. We randomly select 200 Yes/No QA samples. As a result, Pct. Diff reaches 0.04 and FP Ratio achieves 0.40, which reflects limited yes/no bias in the VideoMarathon dataset.
- **Hallucination**: 20 volunteers evaluate the quality of 50 randomly selected OE QA samples. Each sample is rated on a scale from 1 to 10 based on the following criteria. Score 1 denotes a totally irrelevant response, while Score 10 indicates an almost perfect answer with no hallucinations.

Table 7: Human evaluation of hallucination on open-ended QA samples in VideoMarathon dataset.

| Score | 1 | 2 | 3 | 4 | 5 | 6 | 7 | 8 | 9 | 10 |
|---|---|---|---|---|---|---|---|---|---|---|
| **Ratio** (%) | 2.22 | 2.44 | 1.78 | 3.56 | 4.67 | 4.67 | 13.33 | 22.67 | 18.00 | 26.67 |

Higher scores correspond to fewer hallucinations. As show in Table 7, human evaluation achieves an average score of $7.81 \pm 0.59$ (based on ratings from 20 volunteers over 50 OE QA samples), validating the quality of OE QA samples in the VideoMarathon dataset (Scores above 7.5 are considered largely correct). Notably, the score distribution shows that over 80% of the samples receive scores $\geq 7$. To further support the claim "Scores above 7.5 are considered largely correct", we present examples with averaged scores between 7–8 below:

```
================================================================================================
# Example 1
Question: Where is the woman positioned relative to the man in the scene where they are sitting on the
          couch?
Synthetic Answer: The woman is sitting next to the man on the couch.
Ground Truth: The woman is sitting to the right of the man on the couch.
Averaged Score: 8.0

# Example 2
Question: After the boy lights the candle in Clip, what does the woman do?
Synthetic Answer: She holds the candle after it is lit.
Ground Truth: She starts to speak and holds the candle.
Averaged Score: 7.3

# Example 3
Question: What is the woman doing when she uses a sewing machine in the video?
Synthetic Answer: She is sewing the cuffed hem onto the pants.
Ground Truth: She is sewing the hem of the pants.
Averaged Score: 7.5
================================================================================================
```

## A.3    Additional Experimental Details

### A.3.1    Detailed Training Schedules for Hour-LLaVA

In Section 4, we briefly introduce the training schedules of Hour-LLaVA. Furthermore, Table 8 presents the detailed training schedules for each training stage of Hour-LLaVA, containing compression details, data usage, and training hyperparameters.

Table 8: **Detailed training schedule for each training stage of Hour-LLaVA**, including compression details, data usage, and training hyperparameters. In *Data* part, we use the following abbreviations: T for text, SI for single-image, MI for multi-image, SV for short-video, LV for long-video, OV-SI for LLaVA-OV-SI [26], L.V. for LLaVA-Video-178K [70], and V.M. for VideoMarathon.

| | | Image-Language Pretraining | | Video-Language Adaptation | | Video Instruction Tuning | |
|---|---|---|---|---|---|---|---|
| | | **3B** | **7B** | **3B** | **7B** | **3B** | **7B** |
| *Compression* | **FPS** | 1 | 1 | 1 | 1 | 1 | 1 |
| | **Spatial Forgetting (SF)** | 1/4 | 1/4 | 1/4 | 1/4 | 1/4 | 1/4 |
| | SF Mechanism | Random | Random | Random | Random | Random | Random |
| | **Temporal Forgetting (TF)** | - | - | 1/4 | 1/4 | 1/4 | 1/4 |
| | TF Mechanism | - | - | Uniform | Uniform | Uniform | Uniform |
| *Data* | **Data Type** | SI | SI | T + SI + MI + SV | T + SI + MI + SV | T + SI + MI + SV + LV | T + SI + MI + SV + LV |
| | Data Sources | OV-SI | OV-SI | OV-SI + L.V. | OV-SI + L.V. | OV-SI + L.V. + V.M. | OV-SI + L.V. + V.M. |
| | #Samples | 3B | 3B | 0.6M | 0.6M | 4.4M | 4.4M |
| *Training* | **Batch Size** | 128 | 128 | 128 | 128 | 128 | 256 |
| | **LR of Vision Encoder** | 0 | 0 | $5 \times 10^{-6}$ | $5 \times 10^{-6}$ | 0 | 0 |
| | **LR of Projector** | 0 | 0 | $1 \times 10^{-4}$ | $1 \times 10^{-4}$ | $1 \times 10^{-4}$ | $1 \times 10^{-4}$ |
| | **LR of MemAug** | $1 \times 10^{-4}$ | $1 \times 10^{-4}$ | $1 \times 10^{-4}$ | $1 \times 10^{-4}$ | $1 \times 10^{-4}$ | $1 \times 10^{-4}$ |
| | **LR of LLM** | 0 | 0 | $2 \times 10^{-5}$ | $1 \times 10^{-5}$ | $2 \times 10^{-5}$ | $2 \times 10^{-5}$ |
| | **Epoch** | 1 | 1 | 1 | 1 | 1 | 1 |

### A.3.2    Experimental Settings of Ablation Studies.

Due to page limitations, we are unable to include all experimental settings for the ablation studies in Section 5. Below, we provide the experimental settings for each ablation study. We conduct all the ablation studies using the Hour-LLaVA-3B model.

**Ablation Study for VideoMarathon** (Section 5.3). In this section, we investigate how performance changes with different mixtures of VideoMarathon (long-video) and LLaVA-Video-178K (short-video) training data. We construct two subsets from VideoMarathon and LLaVA-Video-178K, matched in both the number of videos and the number of video-language instruction training samples. Each sub-dataset consists of 70K video-language instruction samples over 10K videos. We then train the Hour-LLaVA-3B model on different mixtures of these two subsets (as shown in Figure 3), while also incorporating the same 60K multimodal samples from LLaVA-OV [26] (*i.e.*, 20K text-only, 30K single-image, and 10K multi-image samples) to support multimodal learning. In addition, the Hour-LLaVA-3B model is initialized from the checkpoint obtained after image-language pretraining (Stage 1 in Table 8). All training hyperparameters follow the settings used for *video instruction tuning* of the 3B model, as detailed in Table 8.

**Analysis on Key Components of Hour-LLaVA** (Section 5.4). In this section, we primarily conduct analysis on three key components in Hour-LLaVA: forgetting mechanisms, MemAug module, and memory repository.

We ablate **forgetting mechanisms** from both spatial and temporal perspectives.

- **Spatial Forgetting**. In this setting, we adopt 3B LLaVA-OV-SI data for training. The vision encoder, projector, and LLM decoder of the Hour-LLaVA-3B model are initialized by a pretrained LLaVA-OV-SI-3B model as mentioned in Section 5.1. During training, only the parameters of the MemAug module are tuned, while the rest of the model remains frozen. Training hyperparameters follow the settings used for *image-language pretraining* of the 3B model in Table 8.

- **Temporal Forgetting**. We compare several representative video token compression techniques, including uniform [23, 62], keyframe [45, 50], and question-guided [12, 45] temporal compression. Also, we implement random temporal compression as a reference baseline. In particular, the implementation details of different temporal compression methods are shown below:
  - **Random**. We randomly select $1/4$ frames from a video sampled at 1 FPS.
  - **Uniform**. We uniformly sample $1/4$ frames across the temporal dimension from a 1 FPS video.
  - **Keyframe**. Following [45], for each frame, we compute the average cosine similarity with its $K$ nearest temporal neighbors (with $K = 8$). We then retain the $1/4$ frames that are least similar to their neighbors, removing the $3/4$ most redundant frames. In addition, video frame features are extracted by SigLIP vision encoder [67].
  - **Question-guided**. Following [12], we calculate the cosine similarity between the frame embedding $\mathbf{v}$ and question embedding $\mathbf{q}$. Next, we select $1/4$ frames with the highest similarity scores. In particular, the frame embedding $\mathbf{v}$ is computed as $\mathbf{v} = \frac{1}{N_v} \sum_{i=1}^{N_v} \boldsymbol{v}_i$, where $N_v$ is the number of tokens per frames and $\boldsymbol{v}_i$ refers to the $i$-th visual token vector. The question embedding is calculated as $\mathbf{q} = \frac{1}{N_q} \sum_{i=1}^{N_q} \boldsymbol{q}_i$, where $N_q$ is the number of tokens in the given question and $\boldsymbol{q}_i$ denotes the $i$-th query token vector. Additionally, $\boldsymbol{v}_i$ is obtained using the SigLIP vision encoder followed by the projector, while $\boldsymbol{q}_i$ is derived from the Qwen2.5-3B word embedding model.

  For a fair comparison, all the above temporal compression techniques apply different temporal compression strategies with the same compression ratio of $1/4$ on 1-FPS input videos, along with a random spatial compression with a compression ratio of $1/4$. Note that for training, we use a dataset composed of 70K video-language instruction samples with a 3:1 mixture of VideoMarathon and LLaVA-Video-178K, along with 60K multimodal samples from LLaVA-OV. The Hour-LLaVA-3B model is initialized from the checkpoint obtained after image-language pretraining. All training hyperparameters follow the *video instruction tuning* setup for the 3B model.

For **MemAug module**, we adopt the same training data and recipe as mentioned for ablation studies on forgetting mechanisms. All training hyperparameters are consistent with the *video instruction tuning* configuration of the 3B model in Table 8.

For **memory repository**, we mainly analyze the impact of the memory repository scale in Figure 4 (*middle*). The 100% scale refers to storing full video tokens sampled at 1 FPS. For the 50% and 25% scales, we uniformly retain 50% and 25% of the frame-level features along the temporal dimension, respectively. The <10% scale denotes a lightweight configuration in which only decayed video tokens are replaced in the memory repository, resulting in fewer than 10% of the full video token count. For the training, we follow the same setup as mentioned above. All training hyperparameters are consistent with the *video instruction tuning* configuration of the 3B model in Table 8.

### A.4 Additional Empirical Analysis

**Impact of 1D Structured RoPE**. In MemAug module, we adopt 1D structured RoPE to associate the spatial-temporal correspondence between the full video tokens in the memory repository and the decayed tokens, following prior works [16, 49]. We conduct ablation study on the 1D structured RoPE on *temporal reasoning* subset of VideoMME dataset, as shown in Table 9. Results show a substantial performance drop on the temporal reasoning task when this structural RoPE is removed. It demonstrates that constructing the spatial-temporal association between the full video tokens and decayed video tokens is necessary.

Table 9: Effect of 1D structure RoPE within the MemAug module.

| Method | 1D structure RoPE | VideoMME (Temporal Reasoning) |
|---|---|---|
| Hour-LLaVA-3B | ✓ | 46.3 |
| | × | 35.6 (-10.7) |
| Hour-LLaVA-7B | ✓ | 56.5 |
| | × | 38.4 (-12.1) |

**Performance on Free-form Generation Tasks**. We evaluate Hour-LLaVA-3B and Hour-LLaVA-7B on MLVU-Dev [71] with free-form generation tasks, including sub-science captioning (SSC) and video summarization (SUM). The generated results are measured by GPT-4-Turbo with carefully designed instructions, where the rating score is on a 2–10 scale. We also report results from other state-of-the-art Video-LMMs for reference. In Table 10, the results demonstrate the strong capability of Hour-LLaVA on free-form generation tasks.

Table 10: Performance on free-form generation tasks (MLVU-Dev) evaluated by GPT-4-Turbo. SSC: Sub-Scene Captioning; SUM: Video Summarization.

| Method | Overall | SSC | SUM |
|---|---|---|---|
| MovieChat | 2.78 | 3.23 | 2.33 |
| LLaVA-1.6 | 3.23 | 4.35 | 2.11 |
| ShareGPT4Video-7B | 3.77 | 5.02 | 2.52 |
| LongVA-7B | 4.33 | **5.26** | 3.39 |
| Video-XL-7B | 4.50 | – | – |
| Hour-LLaVA-3B | 4.48 | 5.04 | 3.91 |
| Hour-LLaVA-7B | **4.60** | 5.22 | **3.97** |

**Comparison with recent advanced Video-LMMs**. In Table 11, we present a comparison with the most recent advanced Video-LMMs, including Qwen2-VL [54], Qwen2.5-VL [3], InternVL3 [73], and InternVideo2.5 [56]. Compared among 3B-LMMs, Hour-LLaVA-3B substantially enhances its pretrained model (LLaVA-OV-SI-3B) and is generally comparable to its strong peer Qwen2.5-VL-3B. Compared with the 7B-LMMs, Hour-LLaVA-7B markedly enhances its pretrained model (LLaVA-OV-SI-7B) and surpasses Qwen2-VL-7B. It also performs comparably with, and sometimes outperforms, state-of-the-art models built on the stronger Qwen-2.5-7B LLM decoder.

### A.5 Limitations

Despite the promising results of Hour-LLaVA, several limitations remain. First, due to the lack of comprehensive evaluation metrics for hour-long video-language understanding, multiple-choice QA remains the most practical task for evaluating long video-language models. However, this evaluation format is limited in scope and fails to assess the broader capabilities of Video-LMMs. The development of more diverse and holistic benchmarks is therefore essential for advancing this field. Second, Hour-LLaVA is trained on large-scale synthetic instruction-following data, which inevitably contains noise. The current training pipeline does not explicitly consider this issue, and future work can further explore noise-robust training strategies. Third, the current framework is limited to video and language modalities, neglecting audio, a crucial component in many long-form videos such as lectures, interviews, and documentaries. Incorporating audio or additional modalities could further enhance the model's capacity for comprehensive multimodal understanding.

Table 11: **Performance comparison of existing LMMs** on LongVideoBench, VideoMME, and LVBench datasets. M-Avg denotes the average performance of multiple-choice tasks. The symbol $^\dagger$ marks our reimplemented results. **Bold** font denotes the best performance among models at the same scale, while underline indicates the second-best.

| Method | LongVideoBench | VideoMME (w/o & w/ subtitles) | | | LVBench |
|---|---|---|---|---|---|
| | M-Avg | Overall | Medium | Long | Avg |
| **LLM Decoder: Qwen2.5-3B** | | | | | |
| LLaVA-OV-SI-3B$^\dagger$ [26] | 49.6 | 51.1/54.5 | 49.2/52.1 | 44.1/46.4 | 35.4 |
| Qwen2.5-VL-3B [3] | 54.2 | **61.5/67.6** | 58.7$^\dagger$/**66.3**$^\dagger$ | 51.9$^\dagger$/**61.6**$^\dagger$ | 43.3 |
| Hour-LLaVA-3B (*ours*) | **57.8** | 60.6/66.7 | **59.0**/65.4 | **52.1**/60.4 | **44.7** |
| **LLM Decoder: Qwen2-7B** | | | | | |
| LLaVA-OV-SI-7B [26] | 52.1$^\dagger$ | 58.2/61.5 | 52.6$^\dagger$/55.4$^\dagger$ | 47.9$^\dagger$/50.2$^\dagger$ | 36.0$^\dagger$ |
| Qwen2-VL-7B [54] | 54.6 | 63.3/69.0 | 60.6$^\dagger$/67.8$^\dagger$ | 50.2$^\dagger$/62.7$^\dagger$ | 40.0 |
| Hour-LLaVA-7B (*ours*) | **60.4** | **63.6/70.2** | **63.8/70.0** | **55.0/65.1** | **45.6** |
| **LLM Decoder: Qwen2.5-7B** | | | | | |
| Qwen2.5-VL-7B [3] | 56.0 | 63.3/**69.0** | 62.7$^\dagger$/70.7$^\dagger$ | 53.2$^\dagger$/64.4$^\dagger$ | 45.3 |
| InternVL3-8B [73] | 58.8 | **66.3**/68.9 | -/- | -/- | 40.0 |
| InternVideo2.5-8B [56] | **60.6** | 65.1/- | -/- | -/- | **46.4** |

## A.6 Broader Impacts

This study presents significant advancements in long-form video-language modeling through the introduction of the VideoMarathon dataset and the Hour-LLaVA model. Positively, the VideoMarathon dataset paves the way for more sophisticated AI systems capable of handling realistic, long-duration scenarios, which are essential for practical applications in education, security, autonomous driving, and augmented or virtual reality. However, potential negative impacts include the risk of misuse in surveillance, such as continuous monitoring and profiling of individuals in public or private settings without consent, and the possibility of misinterpreting nuanced or sensitive content in long videos, which might lead to harmful decisions in critical domains like healthcare or security. These implications highlight the importance of responsible development practices, including robust privacy safeguards and clear ethical guidelines for the deployment of long-form video-language models.

## Clip-level Video Captioning

You are given a 30-second video clip. Your task is to generate a detailed, structured description of the video based on six specific perspectives.

### Instructions:

Please ensure your descriptions are vivid, precise, and rich in detail. For each of the six topics below, please provide a comprehensive caption. Your goal is to clearly convey the video's visual and contextual content from multiple dimensions:

- **Temporality**: Describe how the scene evolves over time. Highlight key transitions, unfolding actions, or changes from the beginning to the end of the video.
- **Spatiality**: Describe the spatial layout of the scene. Explain how key elements are positioned and oriented, and how they relate to one another within the frame.
- **Object**: Identify key objects in the video, including inanimate items and humans. Describe their appearance, clothing, physical traits, materials, and possible roles.
- **Action**: Describe the main actions taking place. Specify what actions occur, who or what performs them, and the manner or sequence in which they unfold.
- **Scene**: Provide a high-level overview of the setting. Describe the environment, background, and general activity or context presented in the video.
- **Summary**: Offer a brief yet comprehensive summary that contains the core event or purpose of the given video clip.

### Output Format (JSON):

Your response should be formatted as a JSON object, where each key corresponds to a topic and each value is the description associated with that topic.

#### Example Output:

```json
{
    "Temporality": "The video begins with the group standing idle under the tree and
    progresses to active conversation and movement, suggesting preparation for an
    activity.",
    "Spatiality": "The individuals are grouped under a large tree, spaced out in a
    semicircle. The surrounding area is an open, grassy park with scattered trees in the
    background.",
    "Object": "The scene includes several casually dressed men wearing t-shirts, shorts,
    hats, and sunglasses. One man carries a medium-sized cardboard box. The background
    features natural objects like trees, grass, and park benches.",
    "Action": "The men engage in conversation, gesturing with their hands, and preparing
    for an activity. One man walks away, possibly to retrieve additional items.",
    "Scene": "The video captures a casual outdoor setting in a park. A small group of men
    gather under a shady tree, seemingly preparing for a social or recreational activity
    in a relaxed, natural environment.",
    "Summary": "A group of men gather under a tree in a public park, casually conversing
    and preparing for an activity. The video captures a moment of calm interaction and
    coordination in a relaxed outdoor setting."
}
```

Figure 5: The prompt for clip-level video captioning.

## Event Splitting

You will be given a list of video clips, where each clip is defined by a start time, end time, and its video content. Your task is to merge related clips into a small number of coherent events, each represented by a merged time span and an event title.

### Instructions:

Consider the context for each video clip and ignore some outliers or unreasonable video content. If the given video clips are already an event, do not merge.

- **Merge Related Clips**: Combine consecutive clips that logically belong to the same event based on content continuity.
- **Preserve Context**: Each merged event should represent a self-contained and contextually consistent unit of action or activity.
- **Filter Outliers**: Ignore clips that are irrelevant, inconsistent with the surrounding context, or clearly out of place.
- **Balance Granularity**: Avoid over-segmenting the video. Aim to minimize the number of events while ensuring that each one captures a distinct scene, set of actions, or objects.
- **Respect Standalone Clips**: If a single clip already represents a complete, meaningful event, retain it without merging.

### Output Format (JSON):

Return a JSON object where:
- Each key is a string denoting the merged time span of an event (*e.g.*, "*0–40s*").
- Each value is a concise event title that summarizes the main content or activity.

#### Example Output:

```json
{
    "0-40s": "Introduction to Holiday Desserts",
    "60-110s": "Preparing Ingredients and Tools",
    "120-190s": "Mixing Ingredients",
    "200-290s": "Placing Ingredients into Tools",
    "300-350s": "Baking of Desserts",
    "360-410s": "Presentation and Enjoyment of Desserts",
}
```

Figure 6: The prompt for event splitting.

## Event-level Video Captioning

You will be provided with a sequence of video clips depicting a specific event. Each clip includes metadata (start time, end time) and structured descriptions from six perspectives. Your task is to generate a cohesive, natural-language narrative that summarizes the entire event in chronological order. To support your understanding of the broader context, you will also receive brief descriptions of the events immediately before and after the current one. Use this surrounding context to enhance the flow and interpretation, but do not include those descriptions in your output.

### Instructions:
#### 1. Maintain Chronological Flow
- The clip-level video captions are **already chronologically ordered**. Your summary must **preserve this timeline** and describe the event as a continuous progression.
- **Avoid explicitly referencing individual clips** (*e.g.*, "*The first clip shows*..." or "*As the clip progresses*..."). Instead, describe the event as a seamless and continuous narrative.
- **Do not assume the first or last frame of any clip represents the beginning or end of the entire event.** Instead, focus on how the event unfolds as a whole.

#### 2. Use Adjacent Event Context Thoughtfully
- To **improve coherence and contextual understanding**, you will receive **brief descriptions of the events that occur immediately before and after the current event**.
- Use this information to **better understand** the current event.
- **IMPORTANT:** Do not include these descriptions in your summary. They are provided solely to help you maintain context and continuity.

#### 3. Preserve Key Details, Filter Outliers
- Retain **all relevant details** from the clip descriptions to ensure accuracy and completeness.
- **Based on the brief description of the current event**, ignore **outliers** or clips with inconsistent, irrelevant, or contradictory content that do not contribute meaningfully to the event-level narrative.

#### 4. Write in a Natural, Engaging Tone
- Your summary should **read like a natural video description**, as if you are directly describing the event rather than summarizing segmented clips.
- **Avoid mechanical phrases** often found in clip descriptions, such as "*The clip begins*...", "*As the clip progresses*...", "*The clip concludes*...", "*The first/last frame of this clip*...", "*The second clip shows*...".
- Instead, **focus on crafting a flowing narrative**. The goal is to help a reader visualize the full event as if watching it unfold.

### Output Format (JSON):
The output should be structured as a JSON object.

#### Example Output:
```json
{
    "Event-Level Description": "YOUR DESCRIPTION HERE."
}
```

### Input:
- Brief description of current event: <EVENT TITLE>
- The event before the current event: <PREV EVENT DESCRIPTION>
- The event after the current event: <NEXT EVENT DESCRIPTION>
- Detailed descriptions of all clips in the current event:
    - <CLIP-LEVEL DESCRIPTION 1>
    - <CLIP-LEVEL DESCRIPTION 2>
    - ...
    - <CLIP-LEVEL DESCRIPTION N>

Figure 7: The prompt for event-level video captioning.

## Global-level Video Captioning

You will be provided with a chronologically ordered sequence of video events, each accompanied by both a brief and a detailed description. Your task is to synthesize these descriptions into a single, cohesive, and vivid narrative that summarizes the entire sequence of events as a unified whole, without breaking the flow into separate segments or referencing individual clips mechanically.

### Instructions:
#### 1. Event-level Video Descriptions
Each event is described with two levels of granularity:
- **Brief Description**: A concise overview of the event.
- **Detailed Description**: A more in-depth description, including finer details.

#### 2. Generate a Unified Narrative
- Write a **single, flowing narrative** that describes the full sequence of events from beginning to end.
- Maintain **chronological order** without explicitly mentioning timestamps or referencing individual events (*e.g.*, "*In the second event...*").
- Ensure the narrative reads as if you are describing a continuous experience, **not a series of separate parts**.

#### 3. Focus on Relevance and Consistency
- Incorporate all **meaningful and consistent details** across the event descriptions.
- If a detail appears **inconsistent, irrelevant, or clearly unrelated**, omit it based on your understanding of the overall narrative.

#### 4. Use a Natural and Engaging Tone
- Write in a **fluent, descriptive style**, suitable for someone reading or hearing a natural summary of the video.
- Avoid mechanical phrases like: "*The event begins...*", "*As the event progresses...*", "*The first/last event shows...*".
- Instead, **immerse the reader** in the experience, emphasizing continuity, clarity, and engagement.

### Output Format (JSON):
The output should be structured as a JSON object.

#### Example Output:
```json
{
    "Global-Level Description": "YOUR DESCRIPTION HERE."
}
```

### Input:
Event from <START TIME 1> - <END TIME 1>:
- Brief description: <EVENT TITLE 1>
- Detailed description: <EVENT-LEVEL DESCRIPTION 1>

Event from <START TIME 2> - <END TIME 2>:
- Brief description: <EVENT TITLE 2>
- Detailed description: <EVENT-LEVEL DESCRIPTION 2>

...

Event from <START TIME N> - <END TIME N>:
- Brief description: <EVENT TITLE N>
- Detailed description: <EVENT-LEVEL DESCRIPTION N>

Figure 8: The prompt for global-level video captioning.

## Open-Ended QA Generation

You are an intelligent assistant specializing in **open-ended question-answer generation** for video understanding. Please follow the instructions precisely and **use only the information provided** in the input. Do **not** introduce any content unrelated to the described video clips.

You will be provided with:
- A **chronologically ordered sequence** of <TOPIC>-based video clip descriptions (each 30 seconds long, with start and end timestamps).
- A **global-level description** summarizing the overall content of the video.

Your task is to generate **open-ended Question-Answer (QA) pairs** from the perspective of <TOPIC>. These QA pairs should promote deep understanding and must be **strictly grounded** in the given descriptions. **No hallucination or fabrication is allowed.**

### Instructions:
#### 1. <TOPIC>-Based Sub-Tasks
The <TOPIC>-based sub-tasks can be categorized into the following sub-tasks:
- <SUB-TASK 1>: <TASK DESCRIPTION 1>
- <SUB-TASK 2>: <TASK DESCRIPTION 2>
- *(...additional sub-tasks as needed)*

#### 2. QA Examples for Each Sub-Task
To guide your generation, here are example QA pairs for each sub-task:
Examples of <SUB-TASK 1>:
[
    { "question": <DEMO Q1-1>, "answer": <DEMO A1-1>},
    { "question": <DEMO Q1-2>, "answer": <DEMO A1-2>},
    …
]
Examples of <SUB-TASK 2>:
[
    { "question": <DEMO Q2-1>, "answer": <DEMO A2-1>},
    { "question": <DEMO Q2-2>, "answer": <DEMO A2-2>},
    …
]
…

#### 3. Guidelines for Question-Answer Generation:
- **Focus on <TOPIC>-Relevant Information**: Carefully analyze the descriptions to identify patterns relevant to the <TOPIC>.
- **Relevance and Context**: The questions and answers must align with the content and context of the video clips. The generated question-answer pairs should not introduce information that is not present in the description.
- **Balance Diversity and Clarity**: Create a variety of questions that collectively capture a full understanding of the topic.
- **Quantity**: Generate exactly **three** QA pairs for each sub-task.

### Output Format (JSON):
The output should be structured as a JSON object.

#### Example Output:
```json
{
    "<Sub-Task 1>": [{"question": "<Question 1-1>", "answer": "<Answer 1-1>"}, ...],
    "<Sub-Task 2>": [{"question": "<Question 2-1>", "answer": "<Answer 2-1>"}, ...],
    ...
}
```

### Input:
- <TOPIC>-based Descriptions:
  - Clip from <START TIME 1> - <END TIME 1>: <TOPIC-SPECIFIC CLIP DESCRIPTION 1>
  - Clip from <START TIME 2> - <END TIME 2>: <TOPIC-SPECIFIC CLIP DESCRIPTION 2>
  - …
- Overall Description: <GLOBAL-LEVEL DESCRIPTION>

Figure 9: The prompt for open-ended (OE) question-answer generation.

## Multiple-Choice QA Generation

You are an intelligent assistant specializing in **multiple-choice question-answer generation** for video understanding. Please follow the instructions precisely and **use only the information provided** in the input. Do **not** introduce any content unrelated to the described video clips.

You will be provided with:

- A **chronologically ordered sequence** of <TOPIC>-based video clip descriptions (each 30 seconds long, with start and end timestamps).
- A **global-level description** summarizing the overall content of the video.

Your task is to generate **multiple-choice Question-Option-Answer triplets** from the perspective of <TOPIC>. These triplets should promote deep understanding and must be **strictly grounded** in the given descriptions. **No hallucination or fabrication is allowed.**

### Instructions:
#### 1. <TOPIC>-Based Sub-Tasks
The <TOPIC>-based sub-tasks can be categorized into the following sub-tasks:

- <SUB-TASK 1>: <TASK DESCRIPTION 1>
- <SUB-TASK 2>: <TASK DESCRIPTION 2>
- *(...additional sub-tasks as needed)*

#### 2. QA Examples for Each Sub-Task
To guide your generation, here are example QA pairs for each sub-task:
Examples of <SUB-TASK 1>:
[
    { "question": <DEMO Q1-1>, "options": <DEMO OP1-1>, "answer": <DEMO A1-1>},
    { "question": <DEMO Q1-2>, "options": <DEMO OP1-2>, "answer": <DEMO A1-2>},
    …
]
Examples of <SUB-TASK 2>:
[
    { "question": <DEMO Q2-1>, "options": <DEMO OP2-1>, "answer": <DEMO A2-1>},
    { "question": <DEMO Q2-2>, "options": <DEMO OP2-2>, "answer": <DEMO A2-2>},
    …
]
…

#### 3. Guidelines for Question-Answer Generation:

- **Focus on <TOPIC>-Relevant Information**: Carefully analyze the descriptions to identify patterns relevant to the <TOPIC>.
- **Relevance and Context**: The questions, options, and answers must align with the content and context of the video clips. The generated question-option-answer triplets should not introduce information that is not present in the description.
- **Balance Diversity and Clarity**: Create a variety of questions that collectively capture a full understanding of the topic.
- **Quantity**: Generate exactly **three** QA pairs for each sub-task.

### Output Format (JSON):
The output should be structured as a JSON object.

#### Example Output:
```json
{
  "<Sub-Task 1>": [{"question": "<Question 1-1>", "options": [...], "answer": "<Answer 1-1>"}, ...],
  "<Sub-Task 2>": [{"question": "<Question 2-1>", "options": [...], "answer": "<Answer 2-1>"}, ...],
  ...
}
```

### Input:
- <TOPIC>-based Descriptions:
  - Clip from <START TIME 1> - <END TIME 1>: <TOPIC-SPECIFIC CLIP DESCRIPTION 1>
  - Clip from <START TIME 2> - <END TIME 2>: <TOPIC-SPECIFIC CLIP DESCRIPTION 2>
  - …
- Overall Description: <GLOBAL-LEVEL DESCRIPTION>

Figure 10: The prompt for multiple-choice (MC) question-answer generation.

## Temporality

- **Temporal Perception**: Focuses on recognizing and interpreting the temporal flow of events within a video, including their sequencing, relative timing, and transitions between scenes. The task emphasizes understanding the *local context* of temporal configurations.
- **Duration Perception**: Involves estimating the length of individual events and comparing the durations of multiple actions. The objective is to determine which activities are longer, shorter, or equal in duration. The task emphasizes understanding the *global context* of temporal configurations.
- **Temporal Reasoning**: Requires logical inference based on temporal structure. This includes understanding causal relationships, predicting upcoming events, and reasoning about dependencies between temporally related actions. The task emphasizes understanding the *global context* of temporal configurations.
- **Temporal Gap Estimation**: Entails estimating the elapsed time between two events. The goal is to infer the most accurate time gap based on contextual clues within the video. The task emphasizes understanding the *global context* of temporal configurations.
- **Temporal Order**: Focuses on identifying the correct chronological sequence of events. This task assesses the ability to determine the order in which actions or visual elements appear throughout the video. The task emphasizes understanding the *global context* of temporal configurations.

## Spatiality

- **Spatial Perception**: Focuses on recognizing and interpreting the spatial relationships among objects, people, and movements within a scene. This includes identifying directions, positions, orientations, and local arrangements. The task emphasizes understanding the *local context* of spatial configurations.
- **Spatial Reasoning**: Involves higher-level inference about spatial structures, such as deducing the relative positions, movements, and locations of entities within broader environments. This task requires interpreting interactions across scenes and relies on understanding the *global context* of spatial relationships.

## Object

- **Object Recognition**: Involves identifying and naming specific objects, products, or items that appear in a video. The task also includes providing contextual information about the recognized objects. This task primarily relies on understanding the *local context*.
- **Object Interaction**: Assesses the ability to describe interactions between objects within a scene—how they are used, manipulated, or influence each other. This task emphasizes the *local context* of object dynamics.
- **Object Attribute**: Focuses on describing the visual or functional characteristics of objects, such as appearance, material, color, or size. This task also depends on *local context* understanding.
- **Object Reasoning**: Involves higher-level inference about objects in the scene, such as their roles, functions, or contextual significance. This requires a *global context* understanding across the video.
- **Object Existence**: Evaluates the ability to determine the presence or absence of specific objects at various points in the video. This task requires reasoning over *global context*.
- **Object Permanence**: Assesses the understanding of an object's continuity over time, including its movement, transformation, or disappearance. This task tests the model's ability to track and reason about objects in a *global context*.

## Action

- **Action Recognition**: Focuses on identifying and describing specific actions performed by subjects at given moments in the video. This task requires detailed interpretation of movements and activities within the *local context*.
- **Action Localization**: Aims to determine the temporal and spatial boundaries of actions. It involves identifying when and where actions occur, including their start and end times, and comparing durations. This task relies on understanding the *global context*.
- **Action Sequence**: Involves analyzing the chronological order of actions. The task requires recognizing the sequence of events, including what occurs before or after a given action, and understanding the progression over time. This task requires *global context* reasoning.
- **Action Reasoning**: Requires inferring the motivations or causes behind observed actions. This includes identifying cause-and-effect relationships and interpreting contextual cues to explain why an action was taken. This task depends on *global context* comprehension.

## Scene

- **Scene Recognition**: Assesses the ability to identify and describe the primary setting or environment in which the video takes place (*e.g.*, *indoor vs. outdoor, kitchen vs. park*). This task involves understanding where the events occur and the general atmosphere surrounding the characters. It relies on *global context* comprehension.
- **Scene Transition**: Involves detecting and analyzing changes in scene or setting throughout the video. The task requires recognizing shifts in environment, lighting, background, or thematic focus, and describing these transitions accurately. This task also depends on *global context* understanding.

## Event

- **Event Sequencing**: Involves identifying the correct chronological order of events. This task requires recognizing the sequence in which actions unfold and determining prerequisite steps that lead to specific outcomes. It depends on a coherent understanding of the *global context*.
- **Key Events Extraction**: Focuses on summarizing the most important actions and interactions within a video. The task entails identifying pivotal moments, evaluating their significance, and extracting overarching themes that define the event flow. This also requires comprehensive *global context* comprehension.

Figure 11: Detailed descriptions of 22 sub-tasks over six fundamental topics (except "overall summarization" task). For the "overall summarization" task, we directly use global-level video descriptions as the corresponding answers, so there is no need to design an additional prompt.

```
"20-50s": {
    "Temporality": "The video progresses from the woman introducing the recipe to showing
    the ingredients and possibly demonstrating the steps involved.",
    "Spatiality": "The woman is seated in front of the table, which is positioned against
    a plain wall. The ingredients are arranged neatly on the table.",
    "Object": "The woman wears a black top and a choker necklace with a pendant. On the
    table, there are three bowls containing sugar, flour, and butter, an egg, and a glass
    bowl.",
    "Action": "The woman speaks to the camera, gestures with her hands, and points to the
    ingredients on the table. She appears to be explaining the recipe.",
    "Scene": "The video features a woman sitting in a chair, discussing a recipe while
    showing various ingredients laid out on a table.",
    "Summary": "This video showcases a woman presenting a homemade chocolate salted
    caramel tart recipe. She explains the ingredients and their arrangement on the table,
    providing a step-by-step guide for viewers to follow along."
}
```

(a) Examples of clip-level video captioning.

```
"0-70s": "Introduction and Overview of Chocolate Salted Caramel Tart Recipe",
"60-130s": "Preparing and Kneading the Dough",
"120-190s": "Rolling and Shaping the Tart Crust",
"180-270s": "Creating the Caramel Sauce and Assembling the Tart",
"280-350s": "Finalizing and Presenting the Chocolate Salted Caramel Tart"
```

(b) Examples of event splitting

```
"120-190s": {
    "Brief Description": "Rolling and Shaping the Tart Crust",
    "Detailed Description": "In a cozy kitchen setting, a person meticulously prepares a
    tart crust. The process begins with a ball of dough on a wooden surface, surrounded by
    a generous sprinkling of flour. The individual, dressed in a dark-colored shirt,
    kneads the dough by hand, ensuring it is well-mixed and pliable. Once the dough is
    ready, they use a traditional rolling pin to flatten it, moving the pin back and forth
    to achieve an even thickness. The rolling pin's rhythmic motion is a testament to the
    baker's skill and experience. As the dough is rolled out, it is carefully transferred
    to a round tart pan, which is lined with parchment paper to prevent sticking. The
    person then shapes the dough into the pan, pressing it gently into the sides and
    bottom to create a smooth and even crust. Excess dough is trimmed away, and the final
    touches are added to ensure the tart base is perfect. The entire process is carried
    out with a focused and meticulous approach, highlighting the importance of precision
    in baking. The scene is a blend of traditional techniques and modern kitchen tools,
    creating a harmonious and engaging visual of the tart crust preparation."
}
```

(c) Examples of event-level video captioning.

The video begins with a warm and inviting introduction to a homemade chocolate salted caramel tart recipe. A woman, dressed in a black top and wearing a choker necklace with a pendant, sits in front of a table adorned with colorful candies spelling out 'THE SWEETEST THINGS.' She speaks directly to the camera, gesturing with her hands in a friendly and engaging manner. The scene transitions to her discussing the recipe, pointing out the ingredients laid out on the table: sugar, flour, butter, an egg, and a glass bowl. The setting appears to be a kitchen or dining area, bathed in natural daylight, creating a cozy atmosphere. The woman stands in front of a wooden table, where she begins to mix the ingredients for the tart. Using a fork, she combines the flour, butter, sugar, and egg methodically, demonstrating each step clearly and encouraging viewers to follow along. As the video progresses, the focus shifts to the preparation and kneading of the dough. The woman, now in a slightly messy kitchen environment, works on a wooden table set with all the necessary ingredients. She starts by mixing the flour and butter with a fork, ensuring a thorough blend. After adding an egg and a small amount of water, she continues to mix the dough until it reaches the right consistency. Her hands, now covered in flour, demonstrate a practiced technique as she kneads the dough on the wooden surface, shaping it into a ball. The scene highlights her attention to cleanliness and precision, setting the stage for the next step in the recipe.

The process of rolling and shaping the tart crust follows. The woman, dressed in a dark-colored shirt, works with the dough on a floured wooden surface. She uses a traditional rolling pin to flatten the dough, moving it back and forth to achieve an even thickness. The dough is then carefully transferred to a round tart pan lined with parchment paper. She presses the dough gently into the sides and bottom of the pan, ensuring a smooth and even crust. Excess dough is trimmed away, and the final touches are added to perfect the tart base. The scene showcases a blend of traditional techniques and modern kitchen tools, emphasizing the importance of precision in baking.

Next, the video transitions to the creation of the caramel sauce and the assembly of the tart. The woman places parchment paper into the tart pan and adds rice to weigh it down, ensuring an even distribution of weight. After removing the parchment paper, she applies an egg wash to the pre-baked tart shell. She then prepares the caramel sauce by heating sugar and water in a pot, stirring the mixture with a wooden spoon. Cream and butter are added to create a rich, smooth caramel sauce, which is spread evenly over the tart shell. Dark chocolate chunks are added for flavor and texture, setting the stage for the final steps in the recipe.

The video concludes with the finalization and presentation of the chocolate salted caramel tart. A close-up shows a bowl of melted chocolate being poured into the tart shell, which is placed on a black slate board. The chocolate is spread evenly using a spoon, and the tart is then moved to a wooden cutting board for presentation. Sea salt is sprinkled on top, adding a touch of contrast and enhancing the tart's flavor. The tart is cut into triangular slices with a steady and deliberate technique, ready to be served. The entire process highlights the simplicity and elegance of the chocolate salted caramel tart, from its preparation to its final presentation, leaving viewers inspired to try the recipe themselves.

(d) Examples of global-level video captioning.

Figure 12: Examples of hierarchical video captioning.

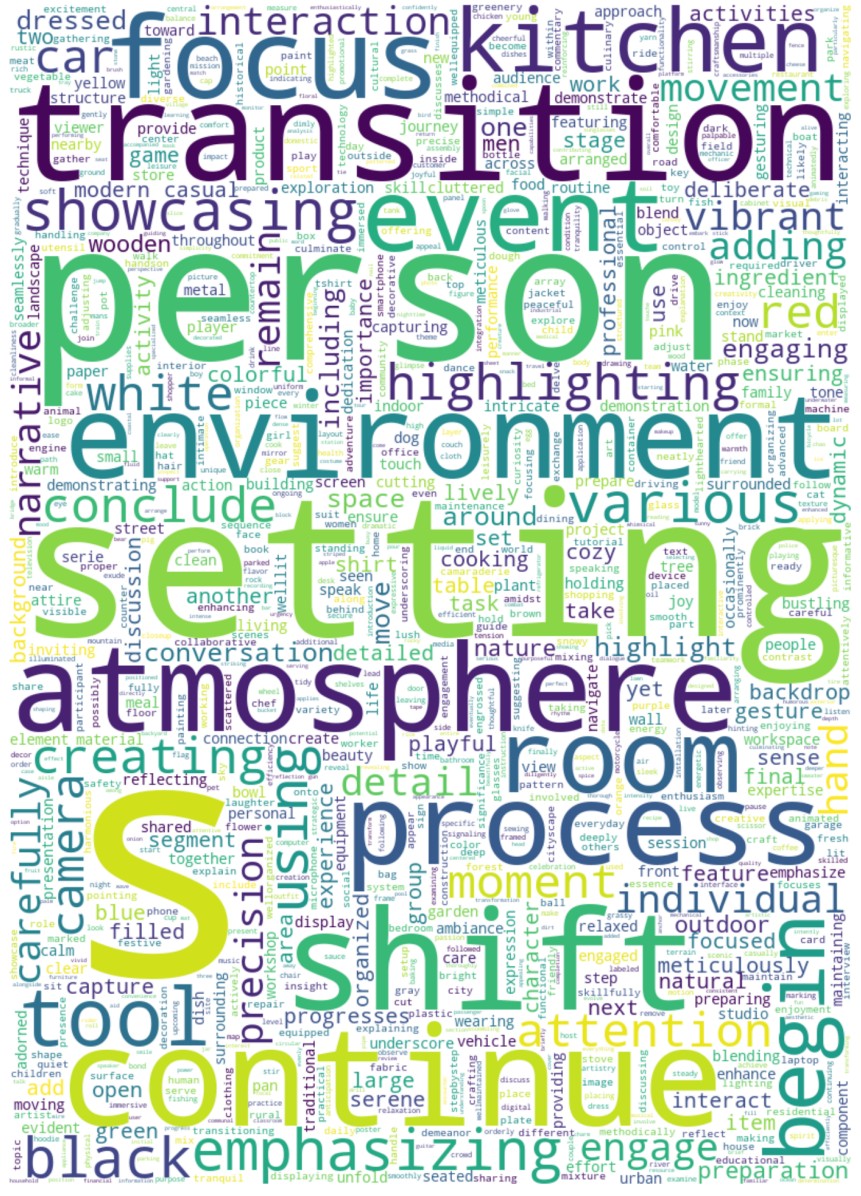

Figure 13: Word cloud of global-level video descriptions in VideoMarathon dataset.

