# OpenReview forum: "Unleashing Hour-Scale Video Training for Long Video-Language Understanding"
_NeurIPS.cc/2025/Conference — NeurIPS 2025 spotlight_

### Official Review · Reviewer_ozeC · 2025-07-01

**Clarity:** 4
**Significance:** 4
**Originality:** 4
**Rating:** 5
**Confidence:** 4

**Summary:**

This paper introduces VideoMarathon, a large-scale, hour-long video instruction-following dataset. It also presents a novel model, Hour-LLaVA, which features three key innovations: a memory repository, a forgetting mechanism, and a MemAug module. Experimental results demonstrate that the combination of Hour-LLaVA and VideoMarathon achieves superior performance on long video benchmarks.

**Questions:**

1. First of all, it would be very interesting if the authors could show some attention heatmaps on the memory repository for a few examples. This would help people understand whether the model learns to retrieve the relevant content.
2. For Weakness 1, could the authors report some performance results on temporal reasoning questions, if possible?

**Ethical Concerns:**

["NO or VERY MINOR ethics concerns only"]

**Final Justification:**

After reading the reviews, rebuttal, and the authors' discussion, I believe the proposed VideoMarathon dataset is a good contribution to the community, and the proposed Hour-LLaVA demonstrates strong results in the comprehensive experiments. Therefore, I will maintain my positive rating.

**Limitations:**

Yes

**Quality:**

4

**Strengths And Weaknesses:**

**Strengths:**

1. The paper is very well-written and easy to follow.
2. The proposed VideoMarathon dataset is a big contribution to the community.
3. The proposed memory repository, forgetting mechanism, and MemAug module seem novel. As lots of works try to compress the video tokens, I strongly agree that we need a memory repository and cross-attention which behaves like retrieval to fetch detailed information if needed.
4. The experimental setup and ablation studies are comprehensive.

**Weakness:**

1. The method uses SigLIP (an image-text model) to extract image features at 1 FPS, then stores these features (tokens) in a memory repository. But these stored features are not temporally processed. The cross-attention would just search for the most relevant content at the frame level, so the model is likely **order-agnostic**, and **temporal relationships may be lost**. I am concerned, especially for tasks that require **temporal reasoning**, such as:
    - “What happened before event X?”
    - “Describe the sequence of actions.”
    - “What changed between minute 5 and minute 10?”

---

> ### Author Rebuttal · Authors · 2025-07-31
>
> Thanks for acknowledging our key contributions. We sincerely appreciate your constructive suggestions, and we hope this response letter can help relieve your remaining concerns.
>
> **W1**: **Temporal relationships may be lost.**
>
> **Ans:** Thanks for raising this concern. In MemAug, we do consider the spatial-temporal relationship between the *full* video tokens in the memory repository and *decayed* video tokens. In particular, we first directly flatten the *full* video tokens into 1D embeddings and apply the 1D structure RoPE, which is commonly used by the prior works [1, 2]. Based on their spatial-temporal relationship, we then construct a corresponding 1D RoPE for *decayed* video tokens. In this way, each *decayed* token (query) is encoded with positional information that matches the similar spatial-temporal positions of its corresponding *full* tokens (keys and values) in the cross-attention module.
>
> Ablation studies show a substantial performance drop on the temporal reasoning task when this structural RoPE is removed. It demonstrates that constructing the spatial-temporal relationship between the *full* video tokens and *decayed* video tokens is necessary.
>
> | Method        | w/ 1D structure RoPE | VideoMME (Temporal Reasoning) |
> | ------------- | -------------------- | ----------------------------- |
> | Hour-LLaVA-3B | Yes                  | 46.3                          |
> |               | No                   | 35.6 (-10.7)                  |
> | Hour-LLaVA-7B | Yes                  | 56.5                          |
> |               | No                   | 38.4 (-12.1)                  |
>
> **Q1**: Attention heatmaps on the memory repository for visualizing whether the model learns to retrieve the relevant content.
>
> **A2**: Thanks for the insightful suggestion. Attention heatmaps over the memory repository indeed offer a way to visualize which frames or regions the model focuses on during retrieval. While we are currently unable to include these visualizations due to policy constraints, we plan to incorporate them in the revised version to better illustrate the model’s behavior.
>
> **Q2**: Results on temporal reasoning tasks.
>
> **Ans**: We evaluate the Hour-LLaVA on VideoMME (Temporal Reasoning subset). We also provide the results of LLaVA-OV-SI and Qwen-VL models as references (* indicates our re-implementation). Notably, Hour-LLaVA achieves considerable improvements in the temporal reasoning task over its pretrained model (LLaVA-OV-SI) with gains of **13.5%** for the 3B model and **20.9%** for the 7B model. Moreover, Hour-LLaVA consistently outperforms the Qwen-VL models on this task, demonstrating its superior temporal reasoning capabilities.
>
> | Method               | VideoMME (Temporal Reasoning) |
> | -------------------- | ----------------------------- |
> | LLaVA-OV-SI-3B       | 32.8                          |
> | Hour-LLaVA-3B (ours) | 46.3 (+13.5)                  |
> | Qwen2.5-VL-3B        | 41.8*                         |
> | LLaVA-OV-SI-7B       | 35.6                          |
> | Hour-LLaVA-7B (ours) | 56.5 (+20.9)                  |
> | Qwen2-VL-7B          | 42.4*                         |
> | Qwen2.5-VL-7B        | 48.5*                         |
>
> **Reference**:
>
> [1] RoFormer: Enhanced transformer with rotary position embedding. Neurocomputing 2024.
>
> [2] TC-LLaVA: Rethinking the transfer from image to video understanding with temporal considerations. ArXiv 2024.

---

> > ### Author Response · Authors · 2025-08-03
> > **Looking Forward to Your Feedback**
> >
> > Dear Reviewer ozeC,
> >
> > It seems we have not received your feedback on our response yet. As the discussion period is nearing its halfway point, we wish to ensure that we have addressed all of your concerns as thoroughly as possible.
> >
> > If there are any remaining questions or points requiring further clarification, we would be most grateful for the opportunity to address them.
> >
> > We sincerely look forward to your feedback. Thanks again for your time!

---

> > > ### Comment · Reviewer_ozeC · 2025-08-04
> > >
> > > Thank you for the detailed responses. I would like to maintain my current rating. I look forward to seeing the visualization and analysis in the revised version.

---

> > > > ### Author Response · Authors · 2025-08-06
> > > > **Official Response by Authors**
> > > >
> > > > Many thanks for your careful reviews and constructive comments! We are glad that our former response has addressed your questions. It has been our pleasure to try out the suggested experiments. Surely, we will add the above empirical results and attention heatmap visualization in the revised version.

---

### Official Review · Reviewer_6Aid · 2025-07-02

**Clarity:** 2
**Significance:** 2
**Originality:** 2
**Rating:** 4
**Confidence:** 5

**Summary:**

The paper introduces VideoMarathon, a large-scale long-video instruction dataset, and Hour-LLaVA, an efficient video-language framework with a memory-augmented module (MemAug) for hour-scale video understanding. VideoMarathon addresses the scarcity of long-video training data by providing 3.3M QA pairs spanning 22 tasks across six themes. The model achieves SOTA results on four long-video benchmarks, demonstrating significant gains over existing video-LMMs.

**Questions:**

1. Could the authors evaluate Hour-LLaVA on generative long-video tasks to validate open-ended reasoning beyond MC-QA?
2. What steps were taken to quantify noise in VideoMarathon’s synthetic QA pairs?
3. Why was N=4 chosen for MemAug blocks?  Would performance improve with deeper architectures, and how does N impact training latency?
4. Can the author add comparisons to Qwen2.5-VL-3B/7B, InternVL3-8B, InternVideo2.5-8B, and LLaVA-OV-SI-3B/7B?

**Ethical Concerns:**

["NO or VERY MINOR ethics concerns only"]

**Final Justification:**

Most of my concerns have been addressed. I am raising my rating to borderline accept.

**Limitations:**

Yes

**Paper Formatting Concerns:**

No major formatting issues were found in the paper.

**Quality:**

3

**Strengths And Weaknesses:**

Strengths
1. VideoMarathon represents a significant advance in long-video instruction data, with 9.7K hours of videos and 3.3M task-oriented QA pairs spanning six thematic categories. The hierarchical captioning pipeline and task-specific QA generation mitigate data scarcity for hour-scale modeling.
2. Extensive experiments across four benchmarks with varying video lengths demonstrate consistent SOTA performance. Ablations validate MemAug’s superiority over heuristic compression methods and VideoMarathon’s necessity for long-context learning .

Weaknesses.
1. Benchmarks exclusively assess multiple-choice QA (Table 2).  Generative capabilities (summarization, causal reasoning)—critical for long-video understanding—remain unevaluated.  Without testing open-ended tasks, claims of holistic "long video-language understanding" are unsubstantiated.
2. All 3.3M QA pairs are synthetically generated via Qwen2VL-7B/DeepSeek-V3.  While prompts include task definitions and examples, no human evaluation or error analysis of QA quality is provided.  Potential noise/ambiguity in generated data may impact model reliability.
3. The paper states MemAug uses N=4 transformer blocks (Sec. 4) but omits ablations on block depth/configurations. The impact of varying N on performance or efficiency remains unverified.
4. Results (Table 2) omit critical baselines like Qwen2.5-VL-3B/7B[1], InternVL3-8B[2], InternVideo2.5-8B[3], and LLaVA-OV-SI-3B/7B. These exclusions weaken claims of superiority, especially given Qwen’s relevance as the backbone for Hour-LLaVA.



[1] Bai S, Chen K, Liu X, et al. Qwen2. 5-vl technical report[J]. arXiv preprint arXiv:2502.13923, 2025.

[2] Chen Z, Wang W, Cao Y, et al. Expanding performance boundaries of open-source multimodal models with model, data, and test-time scaling[J]. arXiv preprint arXiv:2412.05271, 2024.

[3] Wang Y, Li X, Yan Z, et al. InternVideo2. 5: Empowering Video MLLMs with Long and Rich Context Modeling[J]. arXiv preprint arXiv:2501.12386, 2025.

---

> ### Author Rebuttal · Authors · 2025-07-31
>
> We sincerely appreciate your constructive suggestions, and we take these reviews seriously. We hope this response letter can help relieve your concerns.
>
> **W1 & Q1**: Evaluation on open-ended QA benchmarks is needed.
>
> **Ans**: Following your suggestion, we evaluate Hour-LLaVA-3B and Hour-LLaVA-7B on MLVU-Dev [1] with free-form generation tasks, including sub-science captioning (**SSC**) and video summarization (**SUM**). The generated results are measured by GPT-4-Turbo with carefully designed instructions, where the rating score is on a 2–10 scale. We also report results from other state-of-the-art Video-LMMs for reference. The results demonstrate the strong capability of Hour-LLaVA on open-ended tasks.
>
> | Method           | Overall  | SSC  | SUM  |
> | ---------------- | -------- | ---- | ---- |
> | Shargpt4Video-7B | 3.77     | -    | -    |
> | LongVA-7B        | 4.33     | -    | -    |
> | Video-XL-7B      | 4.50     | -    | -    |
> | Hour-LLaVA-3B    | 4.45     | 5.04 | 3.91 |
> | Hour-LLaVA-7B    | **4.57** | 5.22 | 3.97 |
>
> **W2 & Q2**: Human Evaluation is needed.
>
> **Ans**: We agree that the human evaluation on the VideoMarathon dataset is necessary. To quantify the data quality, we manually check 330 randomly selected VideoMarathon QA pairs, including 80 multiple-choice and 250 open-ended samples.
>
> For the *Multiple-Choice (MC) QA* samples, we assess the **Accuracy** of the synthetic samples.
>
> - **Accuracy**: This evaluation assigns a score of 1 if there is exactly one correct answer. A score of 0 is given if the question is irrelevant to the video, if no correct option exists, or if multiple options are correct. The human evaluation on 80 samples achieves an accuracy of **78.7%**, which demonstrates **the reliability of the synthetic multiple-choice QA samples**.
>
> For the *open-ended (OE) QA* samples, we validate the data quality from two perspectives:
>
> - **Yes/No Bias**: Yes/No Bias evaluation [2] validates the extent to which the models tend to respond with ''yes''. In particular, the evaluation consists of ''Yes Percentage Difference'' (Pct. Diff) and ''False Positive Ratio'' (FP Ratio) metrics. Smaller Pct. Diff (**closer to 0.0**) reveals less bias, which means that the predicted number of ''Yes'' is closer to the ground truth. FP Ratio measures how likely the model responds with “Yes” out of all incorrect responses, where a value **closer to 0.5** indicates less bias. We randomly select 200 Yes/No QA samples. As a result, Pct. Diff reaches **0.04** and FP Ratio achieves **0.40**, which reflects **limited yes/no bias in the VideoMarathon dataset**.
>
> - **Hallucination**: 10 volunteers evaluate the quality of 50 randomly selected OE QA samples. Each sample is rated on a scale from 1 to 4 based on the following criteria: **Score 1** - Totally irrelevant; key entities missing; mostly hallucinated; **Score 2** - Key entities present; contains major hallucinations; **Score 3** - Key entities present; contains minor hallucinations; **Score 4** - Almost identical to ground truth; no hallucinations. Scores range from 1 (low quality) to 4 (high quality). Scores above **3.0** are considered *largely correct*. Human evaluation achieves an average score of **3.29 ± 0.27**, indicating that the OE QA samples in the VideoMarathon dataset are consistently reliable.
>
> **W3 & Q3**: Ablation study on the number of MemAug blocks is missing.
>
> **Ans**: Thanks for raising this point. We agree that an ablation study on the number of MemAug blocks is required. Using the experimental setup in Section 5.5, we train four Hour‑LLaVA variants with 1, 2, 4, and 8 MemAug blocks. Across three video benchmarks, performance improves up to 2 or 4 blocks and then decreases slightly at 8. More specifically, for short videos (TempCompass), the sweet spot lies in **the range from 2 to 4 blocks**, while for medium‑ and long‑form videos (LongVideoBench and LVBench), **4 blocks** provides the optimal results.
>
> | # MemAug Blocks | TempCompass | LongVideoBench | LVBench  |
> | --------------- | ----------- | -------------- | -------- |
> | 1               | 59.4        | 53.2           | 40.0     |
> | 2               | **59.8**    | 53.5           | 40.3     |
> | 4               | 59.7        | **54.0**       | **40.6** |
> | 8               | 59.2        | 53.4           | 40.5     |
>
> **W4 & Q4**: Comparisons to advanced Video-LMM are required.
>
> **Ans**: We have listed all the mentioned Video-LMM below. * indicates our re-implementation.
>
> **Comparison among 3B-LMMs**: In this setting, we compare three highly relevant 3B LMMs. LLaVA-OV-SI serves as the pretrained model of Hour-LLaVA, while Qwen-VL and Hour-LLaVA both employ Qwen as their LLM decoder. The results demonstrate that Hour-LLaVA-3B significantly improves the long video-language modeling capability of its pretrained model (LLaVA-OV-SI-3B). Moreover, Hour-LLaVA generally outperforms its strong peer (Qwen2.5-VL-3B), especially in medium- and long-video settings.
>
> | Method         | LLM        | LVBench         | LongVideoBench  | VideoMME (overall) | VideoMME (Medium) | VideoMME (Long) |
> | -------------- | ---------- | --------------- | --------------- | ------------------ | ----------------- | --------------- |
> | LLaVA-OV-SI-3B | Qwen2.5-3B | 35.4            | 49.6            | 51.1               | 49.2              | 44.1            |
> | Hour-LLaVA-3B  | Qwen2.5-3B | **44.7** (+9.3) | **57.8** (+8.2) | 60.6 (+9.5)        | **59.0** (+9.8)   | **52.1** (+8.0) |
> | Qwen2.5-VL-3B  | Qwen2.5-3B | 43.3            | 54.2            | **61.5**           | 58.7*             | 51.9*           |
>
> **Comparison among ~7B-LMMs**: Following prior works [4, 5], we initialize Hour-LLaVA with LLaVA-OV-SI-7B [3], as Qwen-2.5-VL had not yet been released during our training period. Consequently, Hour-LLaVA-7B inherits the **Qwen2-7B** as LLM decoder from its pre-training backbone. For a fair comparison, we also add the results of Qwen2-VL-7B below. Similar to the 3B setting, Hour-LLaVA-7B significantly improves its pretrained model (LLaVA-OV-SI-7B) and surpasses its strong counterpart (Qwen2-VL-7B). Moreover, compared with other state-of-the-art LMMs (Qwen2.5-VL and Intern series) built on the stronger Qwen-2.5-7B backbone, Hour-LLaVA-7B performs comparably on most of the following benchmarks. It even outperforms these models on certain benchmarks, such as the medium and long subsets of VideoMME.
>
> | Method            | LLM        | LVBench     | LongVideoBench | VideoMME (overall) | VideoMME (Medium) | VideoMME (Long) |
> | ----------------- | ---------- | ----------- | -------------- | ------------------ | ----------------- | --------------- |
> | LLaVA-OV-SI-7B    | Qwen2-7B   | 36.0*       | 52.1*          | 58.2               | 52.6*             | 47.9*           |
> | Hour-LLaVA-7B     | Qwen2-7B   | 45.6 (+9.6) | 60.4 (+8.3)    | 63.6 (+5.4)        | **63.8** (+10.2)  | **55.0** (+7.1) |
> | Qwen2-VL-7B       | Qwen2-7B   | 40.0        | 54.6           | 63.3               | 60.6*             | 50.2*           |
> | Qwen2.5-VL-7B     | Qwen2.5-7B | 45.3        | 56.0           | 65.1               | 62.7*             | 53.2*           |
> | InternVL3-8B      | Qwen2.5-7B | -           | 58.8           | **66.3**           | -                 | -               |
> | InternVideo2.5-8B | Qwen2.5-7B | **46.4**    | **60.6**       | 65.1               | -                 | -               |
>
> **Reference**:
>
> [1] MLVU: Benchmarking Multi-task Long Video Understanding. In CVPR 2025.
>
> [2] HallusionBench: An Advanced Diagnostic Suite for Entangled Language Hallucination and Visual Illusion in Large Vision-Language Models. CVPR 2024.
>
> [3] Llava-onevision: Easy visual task transfer. ArXiv 2024.
>
> [4] Longvu: Spatiotemporal adaptive compression for long video-language understanding. ICML 2025.
>
> [5] Long context transfer from language to vision. ArXiv 2024.

---

> > ### Author Response · Authors · 2025-08-03
> > **Looking Forward to Your Feedback**
> >
> > Dear Reviewer 6Aid,
> >
> > It seems we have not received your feedback on our response yet. As the discussion period is nearing its halfway point, we wish to ensure that we have addressed all of your concerns as thoroughly as possible.
> >
> > If there are any remaining questions or points requiring further clarification, we would be most grateful for the opportunity to address them.
> >
> > We sincerely look forward to your feedback. Thanks again for your time!

---

> > ### Comment · Reviewer_6Aid · 2025-08-06
> >
> > Thank you for your detailed responses. While I acknowledge that W3 & Q3 (MemAug block ablation study) has been adequately addressed through the new experiments, several critical concerns regarding other revisions remain unresolved:
> >
> > 1.  W1 & Q1: Incomplete Benchmark Reporting
> >     The SSC and SUM metrics for "other state-of-the-art Video-LMMs" referenced in the Table are still not provided. Furthermore, the calculation methodology for the "Overall" metric is unexplained, making it impossible to assess its validity. Please report all SSC/SUM scores for compared models and define the "Overall" formula.
> >
> > 2.  W2 & Q2: Questionable Hallucination Evaluation
> >     The hallucination score of 3.29 ± 0.27 is unconvincing given the evaluation criteria.  A score of 3 ("minor hallucinations") is deemed acceptable despite admitting inaccuracies, while 4 requires near-perfection. This narrow gap (3 vs. 4) inflates the average and misrepresents reliability.
> >
> > 3.  W4 & Q4: Omission of VideoMME Submetrics
> >     The comparison to advanced Video-LMMs on VideoMME only reports "without sub" scores. The "with sub" metrics are absent. Provide full VideoMME results (with/without sub) for all models.

---

> > > ### Author Response · Authors · 2025-08-06
> > > **Official Response To (W1&Q1 and W4&Q4) by Authors**
> > >
> > > Thank you for participating in the discussion period and for confirming that we have addressed your question on the MemAug block ablation study (**W2&Q2**). We address the remaining question below.
> > >
> > > - **W1 & Q1**: Incomplete Benchmark Reporting
> > >
> > >   **Ans**: Thanks for pointing out the clarity issue. We first provide a clear definition of the "Overall" metric and then present the SSC and SUM results for the compared state-of-the-art Video-LMMs.
> > >
> > >   - **"Overall" Formula**: The MLVU-Dev (Open-ended QA subset) contains 418 test samples, comprising 201 samples for the SSC (sub-scene captioning) task and 217 samples for the SUM (video summarization) task. Model outputs are evaluated by GPT-4-Turbo using carefully designed instructions, with scores assigned on a 2–10 scale. The "Overall" score is computed as the average score over all 418 samples, combining both SSC and SUM tasks. Therefore, the "Overall" metric reflects the model’s overall capability across the two tasks.
> > >
> > >   - **SSC and SUM Results**: In the previous rebuttal, we adopted the results from Video-XL [2], which only reported the “Overall” performance. Upon further review, we found more comprehensive results (including SSC and SUM) in LongVA [1]. The table below reports the SSC and SUM scores for other state-of-the-art Video-LMMs. Hour-LLaVA achieves the leading overall performance, outperforming other methods on the SUM task and delivering competitive performance on the SSC task.
> > >
> > >   All results for other models are taken from LongVA [1] and Video-XL [2]. Because Video-XL-7B does not report SSC and SUM results, the performances on SSC and SUM tasks are left blank.
> > >
> > >   | Method           | Overall  | SSC      | SUM      |
> > >   | ---------------- | -------- | -------- | -------- |
> > >   | MovieChat        | 2.78     | 3.23     | 2.33     |
> > >   | LLaVA-1.6        | 3.23     | 4.35     | 2.11     |
> > >   | Shargpt4Video-7B | 3.77     | 3.77     | 2.52     |
> > >   | LongVA-7B        | 4.33     | **5.26** | 3.39     |
> > >   | Video-XL-7B      | 4.50     | -        | -        |
> > >   | Hour-LLaVA-3B    | 4.45     | 5.04     | 3.91     |
> > >   | Hour-LLaVA-7B    | **4.57** | 5.22     | **3.97** |
> > >
> > >
> > > - **W4 & Q4**: Omission of VideoMME Submetrics
> > >
> > >   **Ans**: We report the results on **VideoMME w/ subtitles** below. * indicates our re-implementation.
> > >   **Comparison among ~7B-LMMs:** Hour-LLaVA-7B markedly enhances its pretrained model (LLaVA-OV-SI-7B) and surpasses Qwen2-VL-7B. It also performs comparably with, and sometimes outperforms, state-of-the-art models built on the stronger Qwen-2.5-7B backbone. To present the comparison more clearly, we separate the Qwen2-7B-based LMMs and Qwen2.5-7B-based LMMs into two tables below.
> > >
> > >   - **Qwen2-7B-based LMMs:**
> > >
> > >     | Method         | LLM      | VideoMME (Overall) | VideoMME (Medium) | VideoMME (Long)   |
> > >     | -------------- | -------- | ------------------ | ----------------- | ----------------- |
> > >     | LLaVA-OV-SI-7B | Qwen2-7B | 58.2/61.5          | 52.6\*/55.4\*     | 47.9\*/50.2\*     |
> > >     | Hour-LLaVA-7B  | Qwen2-7B | **63.6**/**70.2**  | **63.8**/**70.0** | **55.0**/**65.1** |
> > >     | Qwen2-VL-7B    | Qwen2-7B | 63.3/69.0          | 60.6\*/67.8\*     | 50.2\*/62.7\*     |
> > >
> > >   - **Qwen2.5-7B-based LMMs:**
> > >
> > >     | Method            | LLM        | VideoMME (Overall) | VideoMME (Medium) | VideoMME (Long) |
> > >     | ----------------- | ---------- | ------------------ | ----------------- | --------------- |
> > >     | Qwen2.5-VL-7B     | Qwen2.5-7B | 65.1/71.6          | 62.7\*/70.7\*     | 53.2*/64.4\*    |
> > >     | InternVL3-8B      | Qwen2.5-7B | 66.3/68.9          | -                 | -               |
> > >     | InternVideo2.5-8B | Qwen2.5-7B | 65.1/-             | -                 | -               |
> > >
> > >   **Comparison among 3B-LMMs:** Hour-LLaVA-3B substantially enhances its pretrained model (LLaVA-OV-SI-3B) and is generally comparable to Qwen2.5-VL-3B. Here, Qwen2.5-VL-3B achieves better on "w/ subtitles" setting. Given its lower performance in the *"w/o subtitles"* setting but superior results with subtitles, we infer that this advantage might be attributed to the higher quality or larger amount of pure-text data (*not reported*) used in training Qwen2.5-VL, which might enhance its ability to understand video subtitles. The investigation on the impact of pure-text data on Video-LMM performance lies beyond the scope of this work.
> > >
> > >   | Method         | LLM        | VideoMME (Overall) | VideoMME (Medium) | VideoMME (Long)   |
> > >   | -------------- | ---------- | ------------------ | ----------------- | ----------------- |
> > >   | LLaVA-OV-SI-3B | Qwen2.5-3B | 51.1/54.5          | 49.2/52.1         | 44.1/46.4         |
> > >   | Hour-LLaVA-3B  | Qwen2.5-3B | 60.6/66.7          | **59.0**/65.4     | **52.1**/60.4     |
> > >   | Qwen2.5-VL-3B  | Qwen2.5-3B | **61.5**/**67.6**  | 58.7\*/**66.3**\* | 51.9\*/**61.6**\* |
> > >
> > > [1] Long Context Transfer from Language to Vision. arXiv 2024.
> > >
> > > [2] Video-XL: Extra-Long Vision Language Model for Hour-Scale Video Understanding. CVPR 2025.

---

> > > ### Author Response · Authors · 2025-08-06
> > > **Official Response To Hallucination Evaluation (W2&Q2) by Authors**
> > >
> > > **W2 & Q2**: Questionable Hallucination Evaluation
> > >
> > > **Ans**: Thanks for raising concerns about the hallucination evaluation setting. We agree that "the narrow gap (3 vs. 4) inflates the average and misrepresents reliability". To address this, we have refined the scoring criteria into a **1–10 scale**, where Score 1 refers to Totally irrelevant and Score 10 means almost perfect with no hallucinations. A higher score indicates fewer hallucinations.
> > >
> > >   Using this refined scale, human evaluation achieves an average score of **7.81 ± 0.59** (rating by 20 volunteers), validating the quality of OE QA samples in the VideoMarathon dataset (Scores above **7.5** are considered *largely correct*).
> > >
> > >   Furthermore, we report the score distribution to further validate reliability. Notably, over **80%** of the samples receive scores ≥ 7.
> > >
> > >   | Score         | 1    | 2    | 3    | 4    | 5    | 6    | 7     | 8     | 9     | 10    |
> > >   | ------------- | ---- | ---- | ---- | ---- | ---- | ---- | ----- | ----- | ----- | ----- |
> > >   | **Ratio **(%) | 2.22 | 2.44 | 1.78 | 3.56 | 4.67 | 4.67 | 13.33 | 22.67 | 18.00 | 26.67 |
> > >
> > >   To further support the claim "**Scores above 7.5 are considered *largely correct***", we present examples with averaged scores between 7–8:
> > >
> > >   ```
> > >   # Example 1
> > >   Question: Where is the woman positioned relative to the man in the scene where they are sitting on the couch?
> > >   Synthetic Answer: The woman is sitting next to the man on the couch.
> > >   Ground Truth: The woman is sitting to the right of the man on the couch.
> > >   Averaged Score: 8.0
> > >
> > >   # Example 2
> > >   Question: After the boy lights the candle in Clip, what does the woman do?
> > >   Synthetic Answer: She holds the candle after it is lit.
> > >   Ground Truth: She starts to speak and holds the candle.
> > >   Averaged Score: 7.3
> > >
> > >   # Example 3
> > >   Question: What is the woman doing when she uses a sewing machine in the video?
> > >   Synthetic Answer: She is sewing the cuffed hem onto the pants.
> > >   Ground Truth: She is sewing the hem of the pants.
> > >   Averaged Score: 7.5
> > >   ```

---

> > > > ### Author Response · Authors · 2025-08-08
> > > > **Looking Forward to Your Follow-Up Feedback**
> > > >
> > > > Dear Reviewer 6Aid,
> > > >
> > > > Thank you for your thoughtful feedback and for participating in the discussion. We have submitted a further response to each of your remaining concerns and are committed to ensuring every issue is fully resolved.
> > > >
> > > > If any questions or concerns remain, we would be grateful for the opportunity to address them.
> > > >
> > > > Thank you for your time again!

---

> > > > > ### Author Response · Authors · 2025-08-09
> > > > > **Follow-Up: Looking Forward to Your Follow-Up Feedback**
> > > > >
> > > > > Dear Reviewer 6Aid,
> > > > >
> > > > > Thank you for your thoughtful feedback and for participating in the discussion. We have submitted a further response to each of your remaining concerns and are committed to ensuring every issue is fully resolved.
> > > > >
> > > > > If any questions or concerns remain, we would be grateful for the opportunity to address them.
> > > > >
> > > > > Thank you for your time again!

---

> ### Comment · Reviewer_6Aid · 2025-08-09
>
> Thank you for the clarification regarding the "Overall" metric definition and the additional SSC/SUM results. While I appreciate the effort to address the initial concern, significant issues remain unresolved.
>
> The calculation of the "Overall" score (SSC×201/418 + SUM×217/418) directly conflicts with the reported results for multiple models. For instance, SharGPT4Video-7B’s claimed "Overall" score of 3.77 cannot be reconciled with its SSC (3.77) and SUM (2.52) values via the formula, as 3.77×201/418 + 2.52×217/418 ≈ 3.12—a clear 0.65-point discrepancy. Similar inconsistencies exist for other models. This inconsistency between the explicitly stated calculation method and the presented results directly undermines the reliability of the comparative data, regardless of whether the results were sourced from LongVA [1] or Video-XL [2].
>
> [1] Zhang P, Zhang K, Li B, et al. Long context transfer from language to vision[J]. arXiv preprint arXiv:2406.16852, 2024.
>
> [2] Shu Y, Liu Z, Zhang P, et al. Video-xl: Extra-long vision language model for hour-scale video understanding[C]//Proceedings of the Computer Vision and Pattern Recognition Conference. 2025: 26160-26169.

---

> ### Author Response · Authors · 2025-08-09
> **Official Response by Authors**
>
> Thanks for your careful review comments, which have significantly improved the quality of our work.
>
> We acknowledge an unintended misreporting: **ShareGPT4Video-7B’s SSC is 5.02** (not 3.77). After carefully re-examining each row presented in **MLVU [1] (Appendix Table 2) and LongVA [2] (Table 8)**, we confirm that the **"Overall" score is computed as (SSC + SUM)/2**. Accordingly, we have recomputed and corrected the "Overall" scores for Hour-LLaVA-3B and Hour-LLaVA-7B, and the updated table now reflects the accurate values.
>
> | Method           | Overall  | SSC      | SUM      |
> | ---------------- | -------- | -------- | -------- |
> | MovieChat        | 2.78     | 3.23     | 2.33     |
> | LLaVA-1.6        | 3.23     | 4.35     | 2.11     |
> | Shargpt4Video-7B | 3.77     | 5.02     | 2.52     |
> | LongVA-7B        | 4.33     | **5.26** | 3.39     |
> | Video-XL-7B      | 4.50     | -        | -        |
> | Hour-LLaVA-3B    | 4.48     | 5.04     | 3.91     |
> | Hour-LLaVA-7B    | **4.60** | 5.22     | **3.97** |
>
> **Reference:**
>
> [1] MLVU: Multi-task Long Video Understanding Benchmark. CVPR 2025
>
> [2] Long Context Transfer from Language to Vision. arXiv 2024.

---

> > ### Comment · Reviewer_6Aid · 2025-08-09
> >
> > Thank you for the clarification and the additional results. Most of my concerns have been addressed. I am raising my rating to borderline accept.

---

### Official Review · Reviewer_K5fQ · 2025-07-03

**Clarity:** 3
**Significance:** 3
**Originality:** 2
**Rating:** 4
**Confidence:** 3

**Summary:**

This paper presents VideoMarathon, a large-scale hour-long video instruction-following dataset containing approximately 9,700 hours of videos (each 3–60 minutes long) with 3.3M QA pairs across 22 tasks spanning six core topics. Building on this dataset, the paper proposes Hour-LLaVA, a Video-LMM designed for hour-scale video understanding that employs a memory augmentation (MemAug) mechanism to process long videos at 1 FPS. The approach addresses a key gap between short training videos (typically <10 minutes) and long inference videos (>1 hour) in existing models. The proposed method achieves state-of-the-art performance on four video-language benchmarks while maintaining efficiency through learnable compression.

**Questions:**

1. How do you ensure the quality of the 3.3M synthetic QA pairs? What measures were taken to filter hallucinated or inconsistent content from the LMM-generated captions and questions? Was a human evaluation conducted on a subset to validate data quality?
2. The 1/16 compression ratio discards 94% of visual tokens. What information is lost, and have you analyzed how performance degrades with higher compression? What is the theoretical lower bound?
3. For videos exceeding 1 hour, how does the memory repository scale? What is the memory footprint compared to baselines? Can you provide a computational complexity analysis?
4. Given that training videos are up to 60 minutes and LVBench averages 67 minutes (max >2h), how do you explain the generalization? Is this solely due to MemAug, or are there other contributing factors?
5. Why does uniform temporal forgetting outperform more sophisticated strategies like keyframe-based or question-guided selection? Does this suggest that MemAug is doing most of the heavy lifting, making the forgetting mechanism less critical?

**Ethical Concerns:**

["NO or VERY MINOR ethics concerns only"]

**Final Justification:**

Several of my concerns have been addressed. The rebuttal clarifies the importance of MemAug beyond standard cross-attention and token compression in enabling standard Video-LMMs to process hour-scale videos; its integration with the forgetting mechanism is well-motivated and shows consistent gains across token-dropping strategies. The fully synthetic VideoMarathon dataset still carries quality and bias risks, but the human evaluation suggests generally reliable QA pairs, and the paper is transparent about its limitations. The framework is explicitly designed for hour-scale video–language understanding, and the LVBench results demonstrate its effectiveness on long videos. Overall, the contribution is valuable for advancing efficient modeling of long videos, and I lean toward a borderline accept.

**Limitations:**

Yes

**Quality:**

3

**Strengths And Weaknesses:**

Strengths:

1. The work addresses an important problem in video-language understanding — the mismatch between training and inference video lengths.
2. The hierarchical captioning pipeline (clip → event → global) is well-designed and leverages appropriate models (Qwen2VL-7B for captioning, DeepSeek-V3 for summarization).
3. The MemAug mechanism is technically sound, using transformer blocks with cross-attention to recover information from compressed representations.
4. The paper is well-written with clear motivation and methodology. Figures effectively illustrate the architecture and dataset pipeline.
5. Experiments are comprehensive, covering multiple benchmarks with diverse video lengths.

Weaknesses:
1. The core components are mostly adaptations of existing techniques:
(a) MemAug is essentially standard transformer blocks with cross-attention, similar to retrieval-augmented architectures.
(b) The forgetting mechanism is a standard spatial/temporal token reduction at a 1/4 ratio.
(c) Hierarchical captioning follows standard LMM practices across multiple granularities.
2. Heavy reliance on synthetic data generation using LMMs may introduce systematic biases and quality inconsistencies.
3. The paper lacks analysis of noise in the 3.3M QA pairs or quality control measures beyond prompt design.
4. The 22-task taxonomy, while comprehensive, appears somewhat arbitrary without theoretical justification for these particular categories.

---

> ### Author Rebuttal · Authors · 2025-07-31
>
> Thanks for the constructive comments. We hope this response letter can help relieve your concerns.
>
> **W1**: Methodology is mostly adaptations of existing techniques, such as the cross-attention module and standard spatial/temporal token reduction. Hierarchical captioning follows standard LMM practices.
>
> **A1**: Thanks for raising this concern. As cross-attention and self-attention modules are fundamental building blocks of modern LLMs, their presence is not sufficient to determine whether the overall framework is novel or not. To clarify the **novelty of our Hour-LLaVA framework**, we analyze the limitations of existing long video large multimodal models (Video-LMMs), explain the necessity of the Hour-LLaVA framework, and **provide additional empirical results** to support our claims below.
>
> - **Limitations of existing long Video-LMMs**: Existing long Video-LMMs confront a fundamental dilemma:
>   - *Compression*-based methods prioritize computation affordable and training efficiency, but inevitably suffer from **information loss** caused by aggressive compression (e.g., uniform sampling 64 frames from a one-hour video);
>   - *Extension*-based methods preserve high-fidelity video context by using sequence parallel (SP) at the cost of **significantly increased training time and GPU usage** (linear growth with video length), particularly for hour-scale videos.
>
> - **Necessity of Hour-LLaVA framework:** To break this dilemma, we propose the Hour-LLaVA framework, which intends to keep **high-fidelity video context** while prioritizing **training efficiency.** Beyond the extension-based methods, Hour-LLaVA introduces the **forgetting mechanism** (implemented by spatial-temporal token dropping) that decouples computation from the input sequence length, so training and inference costs no longer rise linearly with video length. **MemAug module** (implemented by a cross-attention mechanism) is designed to dynamically recover lost video tokens from the **memory repository**. Without the MemAug module, purely compression‑based approaches inevitably incur irreversible context loss, which leads to sub-optimal performance. As a result, by coupling the forgetting mechanism with MemAug, Hour‑LLaVA delivers compression‑level efficiency while still preserving the full video context sought by extension methods.
>
> - **Hour‑LLaVA is more than a cross-attention module**: The novelty of Hour-LLaVA lies in the *interaction* of the two tightly coupled components: (1) the forgetting mechanism guarantees training efficiency, and (2) MemAug adaptively re-injects video tokens from the memory repository to recover information loss caused by the forgetting mechanism. However, MemAug (implemented by cross attention) is necessary but not sufficient. Without the forgetting mechanism, the trade‑off between contextual fidelity and training efficiency collapses.
>
> - **Hour‑LLaVA does more than frames/tokens dropping**: its learnable **MemAug** module *re‑injects* informative cues that token dropping would otherwise ignore. Moreover, the ablation study below shows that the MemAug module consistently improves the performance across different token dropping strategies, which highlights that MemAug effectively recovers information lost during compression and enhances the overall video-language modeling.
>
>   | Token Dropping Strategy | MemAug | LongVideoBench  | LVBench         |
>   | ----------------------- | ------ | --------------- | --------------- |
>   | Uniform                 | w/     | 54.0  | 40.6  |
>   |                         | w/o    | 52.1            | 38.3            |
>   | Random                  | w/     | 52.2  | 38.8  |
>   |                         | w/o    | 51.7            | 38.1            |
>   | Keyframe                | w/     | 53.5 | 40.3|
>   |                         | w/o    | 52.0            | 38.9            |
>   | Question-Guided         | w/     | 53.4  | 39.7  |
>   |                         | w/o    | 51.0            | 37.5            |
>
> **W2**: Heavy reliance on synthetic data generation is unreliable.
>
> **A3**: We agree that heavily relying on synthetic data is risky. However, human annotation is too expensive to keep pace with the rapidly expanding video data, especially for long videos. Recent advances in LMMs (e.g., GPT-4o and DeepSeek-V3)  allow synthetic data to approach or even surpass human quality. In this way, a virtuous cycle is created, which means better LMMs produce higher-quality synthetic data, which in turn further improves subsequent models. Moreover, empirical studies consistently show that incorporating carefully constructed synthetic data improves the performance of video-LMMs, highlighting the practical value of synthetic data.
>
> **W3 & Q1:** Human evaluation is required.
>
> **A4**: We agree that the human evaluation on the VideoMarathon dataset is necessary. To quantify the data quality, we manually check 330 randomly selected VideoMarathon QA pairs, including 80 multiple-choice and 250 open-ended samples.
>
> For the *Multiple-Choice (MC) QA* samples, we assess the **accuracy** of the synthetic samples by assigning a score of 1 if there is exactly one correct answer. A score of 0 is given if the question is irrelevant to the video, if no correct option exists, or if multiple options are correct. The human evaluation on 80 samples achieves an accuracy of **78.7%**, which demonstrates **the reliability of the synthetic multiple-choice QA samples**.
>
> For the *open-ended (OE) QA* samples, we validate the data quality from two perspectives:
>
> - **Yes/No Bias**: Yes/No Bias evaluation [1] validates the extent to which the models tend to respond with ''yes''. In particular, the evaluation consists of ''Yes Percentage Difference'' (Pct. Diff) and ''False Positive Ratio'' (FP Ratio) metrics. Smaller Pct. Diff (**closer to 0.0**) reveals less bias, which means that the predicted number of ''Yes'' is closer to the ground truth. FP Ratio measures how likely the model responds with “Yes” out of all incorrect responses, where a value **closer to 0.5** indicates less bias. We randomly select 200 Yes/No QA samples. As a result, Pct. Diff reaches **0.04** and FP Ratio achieves **0.40**, which reflects **limited yes/no bias in the VideoMarathon dataset**.
> - **Hallucination**: 10 volunteers evaluate the quality of 50 randomly selected OE QA samples. Each sample is rated on a scale from 1 to 4 based on the following criteria. Scores range from 1 (low quality) to 4 (high quality). Scores above **3.0** are considered *largely correct*. Human evaluation achieves an average score of **3.29 ± 0.25**, indicating that the OE QA samples in the VideoMarathon dataset are consistently reliable.
>
> **W4:** The 22-task taxonomy is comprehensive but somewhat arbitrary.
>
> **A5**: We need to clarify that the 22-task taxonomy in the VideoMarathon dataset is not arbitrary but carefully designed based on an extensive survey of existing video-language datasets (Line 96). These datasets collectively cover over 100 distinct tasks. The current 22-task taxonomy is a distilled summary of those tasks across a wide range of benchmarks, aiming to provide a comprehensive and structured categorization of video-language understanding tasks.
>
> **Q2**: Analysis on **higher compression ratios.** What is the theoretical lower bound?
>
> **A6**: Thanks for the comment. It is challenging to provide a theoretical lower bound for the video token compression ratio due to the complexity of video modeling. To address this concern, we provide an empirical analysis to identify a suitable compression range. As shown in **Figure 4 (left)**, we evaluate the performance under higher temporal compression ratios. The results indicate that a ratio of **1/2 to 1/4** achieves a great balance between computational efficiency and performance. With a compression ratio beyond 1/4, performance begins to degrade significantly.
>
> **Q3.1**: How does the memory repository scale? What is the memory footprint compared to baselines?
>
> **A7**: MemAug with memory repository is a light-weight module. We compare the memory usage of Hour-LLaVA with its baseline when processing different numbers of frames during inference. We report the additional memory usage below, which demonstrates the efficiency of Hour-LLaVA.
>
> | # Frames | Additional Memory Usage |
> | -------- | -------------------------- |
> | 1,800    | +8%                       |
> | 3,600    | +13%                       |
> | 5,400    | +23%                       |
> | 7,200    | +40%                      |
>
> **Q3.2**: A complexity comparison between Hour-LLaVA and its base model is required.
>
> **A9**: **Figure 4 (right)** compares the number of visual tokens fed into the LLM decoder across Hour-LLaVA, LLaVA-Video, and vanilla 1-FPS sampling methods. The larger token number indicates the higher computational complexity, highlighting the low complexity of Hour-LLaVA.
>
> **Q4**: How to explain the generalization of Hour-LLaVA for videos longer than the training period? Are there other contributing factors?
>
> **A10**: The light-weight MemAug module allows Hour-LLaVA to process the longer videos (more than 2 hours). The proposed VideoMarathon dataset serves as another key contributing factor to the generalization of Hour-LLaVA.
>
> **Q5**: Explanation of the superiority of uniform temporal forgetting. Is the MemAug module responsible for most of the heavy lifting?
>
> **A11**: The table in **A1** shows that uniform temporal forgetting performs comparably to keyframe-based forgetting. Although we report different token compression strategies, developing a more advanced token compression strategy with clear superiority is not the focus of this work.
> The table in **A1** also presents that when combined with MemAug, most token compression methods achieve a performance gain of approximately **1.5%–2.5%**, suggesting that the MemAug module plays a major role in boosting overall performance. This finding may serve as supporting evidence for your question.

---

> > ### Author Response · Authors · 2025-08-03
> > **Looking Forward to Your Feedback**
> >
> > Dear Reviewer K5fQ,
> >
> > It seems we have not received your feedback on our response yet. As the discussion period is nearing its halfway point, we wish to ensure that we have addressed all of your concerns as thoroughly as possible.
> >
> > If there are any remaining questions or points requiring further clarification, we would be most grateful for the opportunity to address them.
> >
> > We sincerely look forward to your feedback. Thanks again for your time!

---

> > ### Comment · Area_Chair_XrZk · 2025-08-06
> > **Re: Rebuttal by Authors**
> >
> > Dear Reviewer K5fQ,
> >
> > could you please let us know your feedback with respect to the authors rebuttal?
> >
> > Best!
> > Your friendly AC

---

> > > ### Author Response · Authors · 2025-08-07
> > > **Follow Up: Looking Forward to Your Feedback**
> > >
> > > Dear Reviewer K5fQ,
> > >
> > > It seems we have not received your feedback on our response yet. As the discussion period will end soon, we were wondering if there is anything else we could address to help further clarify? We sincerely look forward to your feedback. Thanks again for your time!

---

> > > > ### Comment · Reviewer_K5fQ · 2025-08-08
> > > > **Response to Rebuttal**
> > > >
> > > > Thank you for the rebuttal and additional experiments. While I appreciate the clarifications, several of my earlier concerns remain only partially addressed:
> > > > 1. Novelty & Technical Contribution - The explanation that Hour-LLaVA’s novelty lies in coupling the forgetting mechanism with MemAug is reasonable, but the rebuttal does not fully resolve the impression that both components are standard adaptations. The empirical gains (≈1.5–2.5%) over different token-dropping strategies suggest MemAug is doing most of the heavy lifting.
> > > > 2. Synthetic Data Quality & Bias - The dataset is entirely generated by LMMs, which brings risks of bias and noise. The provided human evaluation helps, but the subset size (330 out of 3.3M) is small, and the evaluation criteria may still inflate the perceived quality (especially given that “minor hallucinations” count as largely correct). It remains unclear how much systematic bias, style artifacts, or domain imbalance exists in the synthetic data, and whether these impact downstream generalization.
> > > > 3. Generalization Beyond Training Length - The rebuttal attributes generalization to longer videos mainly to MemAug and dataset quality, but this is asserted rather than rigorously demonstrated.
> > > > 4. Forgetting Mechanism Role - While uniform temporal forgetting works comparably to more “sophisticated” strategies, this could imply that the forgetting stage is not critical.
> > > >
> > > > The rebuttal clarifies some aspects, the paper still does not clearly demonstrate the novelty of MemAug beyond adaptations of standard cross-attention and token reduction methods. The reliance on a fully synthetic dataset generated by LMMs leaves open questions on data quality, potential bias, and noise, with the limited human evaluation insufficient to fully validate the 3.3M QA pairs. The explanations for generalization to much longer videos and for the efficiency trade-offs remain high-level and not fully convincing.

---

> > > > > ### Author Response · Authors · 2025-08-09
> > > > > **Official Response by Authors (1/2)**
> > > > >
> > > > > Thank you for participating in the discussion period. We address the remaining question below.
> > > > >
> > > > > - **Novelty & Technical Contribution**
> > > > >
> > > > >   **Ans**: The novelty of this work lies in two perspectives:
> > > > >
> > > > >   - **Dataset - VideoMarathon**: We introduce VideoMarathon, a large-scale hour-long video instruction-following dataset. Compared with the existing video instruction-following datasets [1,2,3], it offers a significantly longer average video length, broader duration range, and a larger number of QA pairs.
> > > > >   - **Methodology - Hour-LLaVA**: Even with available scaled hour-long training data, **naively training existing Video-LMMs on such long training data remains non-trivial** since hour-scale videos explode token budgets  (quadratic attention). Token compression is a straightforward way to alleviate computational cost and improve training efficiency. However, compression-based methods introduce a new issue of information loss. To this end, we propose Hour-LLaVA, which achieves a trade-off between **training efficiency** and **high-fidelity video context**, enabling efficient learning on hour-long videos while reducing the loss of critical context.
> > > > >     - The **MemAug module** is one of the key components within the Hour-LLaVA framework. The purpose of MemAug is to supplement compressed/decayed visual tokens with recovered cues to mitigate information loss. The phenomenon of "The empirical gains (≈1.5–2.5%) over different token-dropping strategies suggest MemAug is doing most of the heavy lifting." highlights **the effectiveness and generalization of the MemAug** towards different "token-dropping strategies".
> > > > >
> > > > > - **Synthetic Data Quality & Bias**
> > > > >
> > > > >   **Ans**: Thank you for raising these important points. We need to clarify that the **purpose** of human evaluation on synthetic data is *not to demonstrate that the data is hallucination/bias-free*, but to show that *hour-scale video-language training is feasible and beneficial* even when training data contains minor hallucination and bias.
> > > > >
> > > > >   - On the evaluation subset size (330 out of 3.3M): A 330-sample, 20-volunteer review aligns with community practice and even exceeds common setups. For instance, ShareGPT4Video [1] uses a 100-sample validation set rated by 10 volunteers. Expanding human review to a majority of 3.3M samples would lead to unaffordable human labor and would deviate from the purpose and advantages (e.g., low-cost scalability) of synthetic data generation.
> > > > >   - Inflation Issue: We have updated a more fine-grained 1–10 rating scale in the general response. Score 1 denotes a totally irrelevant response, while Score 10 indicates an almost perfect answer with no hallucinations. Higher scores correspond to fewer hallucinations. Using this more fine-grained scale, human evaluation achieves an average score of **7.81 ± 0.59** (based on ratings from 20 volunteers over 50 OE QA samples), validating the quality of OE QA samples in the VideoMarathon dataset (Scores above **7.5** are considered *largely correct*). Furthermore, we report the score distribution to further validate reliability. Notably, over **80%** of the samples receive scores ≥ 7. To further support the claim "Scores above 7.5 are considered **largely correct**", we present examples with averaged scores between 7–8. Please refer to the general response: *Updates on Hallucination Evaluation for Open-Ended QA Samples*.
> > > > >   - Uncertainty of systematic bias, style artifacts, and domain imbalance: The systematic bias and style artifacts are inherent limitations of current LLMs and are beyond the scope of this work, which focuses on enabling hour-scale video-language training. For the domain imbalance issue, we have visualized the distributions of the data source, question types, and video duration of the VideoMarathon dataset in Fig. 1(b-d), which demonstrates diverse coverage and balanced mix across video domains and question types.
> > > > >
> > > > > [**Remaining responses are on the next page..**]

---

> > > > > > ### Author Response · Authors · 2025-08-09
> > > > > > **Official Response by Authors (2/2)**
> > > > > >
> > > > > > - **Generalization Beyond Training Length**
> > > > > >
> > > > > >   **Ans**: Thanks for raising this point. To further demonstrate the generalization beyond training length, we compare three models on LVBench (hour-long video-language benchmarks with an average duration of 2466s) below. LLaVA-OV-SI-7B [4] is an image-only pretrained baseline used to initialize both LLaVA-Video-7B and Hour-LLaVA-7B. LLaVA-Video-7B [3] is trained on videos <10 minutes with fixed-length uniform temporal sampling, whereas Hour-LLaVA-7B is trained on hour-scale videos and uses MemAug for video context enhancement. LLaVA-Video-7B (trained on short videos) improves over the image-only baseline, demonstrating that current LMMs inherently can generalize to longer videos. Hour-LLaVA-7B, trained on hour-scale videos and equipped with MemAug for context enhancement, achieves a further improvement on LVBench, demonstrating that hour-scale training combined with MemAug further strengthens video length generalization.
> > > > > >
> > > > > >   | Method                            | LVBench  |
> > > > > >   | --------------------------------- | -------- |
> > > > > >   | LLaVA-OV-SI-7B (Pretrained Model) | 36.0     |
> > > > > >   | LLaVA-Video-7B                    | 42.2     |
> > > > > >   | Hour-LLaVA-7B (ours)              | **45.6** |
> > > > > >
> > > > > > - **Forgetting stage is not critical**
> > > > > >
> > > > > >   **Ans**: We need to clarify the misunderstanding of the forgetting mechanism.  The purpose of the forgetting mechanism is to compress the amount of visual tokens into an affordable number, enabling the LLM decoder to operate directly on fewer tokens. Without a forgetting mechanism, training current Video-LMMs on hour-scale videos at 1-FPS is computationally infeasible.
> > > > > >
> > > > > > **Reference:**
> > > > > >
> > > > > > [1] ShareGPT4Video: Improving Video Understanding and Generation with Better Captions. NeurIPS 2024.
> > > > > >
> > > > > > [2] Direct Preference Optimization of Video Large Multimodal Models from Language Model Reward. NAACL 2025.
> > > > > >
> > > > > > [3] Video Instruction Tuning with Synthetic Data. arXiv 2024.
> > > > > >
> > > > > > [4] LLaVA-OneVision: Easy Visual Task Transfer. arXiv 2024.

---

### Official Review · Reviewer_6t1N · 2025-07-04

**Clarity:** 3
**Significance:** 3
**Originality:** 2
**Rating:** 4
**Confidence:** 4

**Summary:**

This paper introduces a new framework for training video-language models on hour-long videos instead of the usual short clips. They address the scalability and memory issues that come with processing such long content.

Key Contributions:

1. Present **VideoMarathon**, a large-scale hour-long video instruction-following dataset (sources are Panda-70M [7], Ego4D [15], ActivityNet [3], YouCook2 [67], and MovieChat-1K [44]).
    - They achieve this through hierarchical video captioning (captioning segments and incrementally fusing dense captions together),
        1. CLIP level: leverages Qwen2VL-7B LMM as the video clip captioner.
        1. Fuses captions into "event level" (using deepseekv3).
        1. Fuses captions into global level (using deepseekv3).
2. **HourLLAVA:** The manuscript also introduces HourLLAVA , a Video-LMM capable of processing hour-long videos at 1 FPS in both training and inference.
    - They design a memory augmentation mechanism to enable hour-scale video-language understanding. This mechanism consists of three main components:
    - a memory repository (the high-fidelity video features Hv, extracted at 1 FPS w/ SigLIP)
    - a forgetting mechanism (compresses the full video tokens by discarding tokens).
        - Randomly drops 3/4 of the tokens spatially, and uniformly drops 3/4 of the frames temporally.
    - a MemAug module (cross attention to recover information from the full memory repository - ie the uncompressed video features).

Results show that HourLLAVA trained on VideoMarathon outperforms models like Llava-Video.

**Questions:**

Have the authors validated the LLM-synthesised QA pairs for correctness or bias, or measured hallucination rates?

**Ethical Concerns:**

["NO or VERY MINOR ethics concerns only"]

**Final Justification:**

I switched my score to Accept after seeing the rebuttals.
My original review criticized the missing evaluation of the quality of the generated samples. Their added evaluation does show promise, although more could be done (eg more than 80 MC questions evaluated).

Furthermore, the added comparisons and results do strengthen their architecture contribution, despite being simple (this simplicity might even be a strength then). And the good results are achieved with 1/16th~6% of the tokens.

**Limitations:**

Yes, in supplement.

**Quality:**

3

**Strengths And Weaknesses:**

VideoMarathon contributes almost 10k hours of videos with captions and QA pairs. Most existing long video datasets are small and useful mostly for evaluation (eg EgoSchema, HourVideo), so this contribution will help close the training–inference length gap.

Architecture achieves good results with 1/16th~6% of the tokens.

Weaknesses:
- Missing human validation of the questions and answers generated (or at least some analysis of what the hallucination rate is).
- The contribution is minor in that they mostly just adapt existing MLLMs by using heuristics for dropping frames. The only learnable component is the cross attention based MemAug module. This module by itself is not novel and resembles most of the literature on long video understanding - cross attention is a standard technique to avoid the quadratic explosion of self attention layers when inputs are long video sequences.  Furthermore, these references are missing (neither cited nor results referenced). Eg. Vamba combines CA and mamba layers and outperforms HourLlava on LVBench.

---

> ### Author Rebuttal · Authors · 2025-07-31
>
> We sincerely appreciate your constructive suggestions, and we take these reviews seriously. We hope this response letter can help relieve your concerns.
>
> **W1 & Q1**: Human Evaluation on the Synthetic VideoMarathon Dataset.
>
> **A1**: Thank you for highlighting the need to validate the quality of synthetic data. We agree that the human evaluation on the VideoMarathon dataset is necessary. To quantify the data quality, we manually check 330 randomly selected VideoMarathon QA pairs, including 80 multiple-choice and 250 open-ended samples.
>
> For the *Multiple-Choice (MC) QA* samples, we assess the **accuracy** of the synthetic samples by assigning a score of 1 if there is exactly one correct answer. A score of 0 is given if the question is irrelevant to the video, if no correct option exists, or if multiple options are correct. The human evaluation on 80 samples achieves an accuracy of **78.7%**, which demonstrates **the reliability of the synthetic multiple-choice QA samples**.
>
> For the *open-ended (OE) QA* samples, we validate the data quality from two perspectives:
>
> - **Yes/No Bias**: Yes/No Bias evaluation [1] validates the extent to which the models tend to respond with ''yes''. In particular, the evaluation consists of ''Yes Percentage Difference'' (Pct. Diff) and ''False Positive Ratio'' (FP Ratio) metrics. Smaller Pct. Diff (**closer to 0.0**) reveals less bias, which means that the predicted number of ''Yes'' is closer to the ground truth. FP Ratio measures how likely the model responds with “Yes” out of all incorrect responses, where a value **closer to 0.5** indicates less bias. We randomly select 200 Yes/No QA samples. As a result, Pct. Diff reaches **0.04** and FP Ratio achieves **0.40**, which reflects **limited yes/no bias in the VideoMarathon dataset**.
> - **Hallucination**: 10 volunteers evaluate the quality of 50 randomly selected OE QA samples. Each sample is rated on a scale from 1 to 4 based on the following criteria: **Score 1 **- Totally irrelevant; key entities missing; mostly hallucinated; **Score 2** - Key entities present; contains major hallucinations; **Score 3** - Key entities present; contains minor hallucinations; **Score 4** - Almost identical to ground truth; no hallucinations. Scores range from 1 (low quality) to 4 (high quality). Scores above **3.0** are considered *largely correct*. Human evaluation achieves an average score of **3.29 ± 0.25**, indicating that the **OE QA samples in the VideoMarathon dataset are consistently reliable**.
>
> **W2.1**: Limited novelty of the proposed method.
>
> **A2.1:** Thanks for raising concerns about the novelty of the proposed method. As cross-attention and self-attention modules are fundamental building blocks of modern LLMs, their presence is not sufficient to determine whether the overall framework is novel or not. To clarify the **novelty of our Hour-LLaVA framework**, we analyze the limitations of existing long video large multimodal models (Video-LMMs), explain the necessity of the Hour-LLaVA framework, and **provide additional empirical results** to support our claims below.
>
> - **Limitations of existing long Video-LMMs**: Existing long Video-LMMs confront a fundamental dilemma:
>   - *Compression*-based methods prioritize computation affordable and training efficiency, but inevitably suffer from **information loss** caused by aggressive compression (e.g., uniform sampling 64 frames from a one-hour video);
>   - *Extension*-based methods preserve high-fidelity video context by using sequence parallel (SP) at the cost of **significantly increased training time and GPU usage** (linear growth with video length), particularly for hour-scale videos.
>   - As video length becomes longer (e.g., introducing VideoMarathon with hour-scale videos), such a dilemma will be intensified. Compression-based methods must incur greater information loss, while *extension*-based methods should tolerate longer training/inference time. Therefore, a new framework that effectively balances **video context** with **training efficiency** is urgently needed.
>
> - **Necessity of Hour-LLaVA framework:** To break this dilemma, we propose the Hour-LLaVA framework, which intends to keep **high-fidelity video context** while prioritizing **training efficiency.** Beyond the extension-based methods, Hour-LLaVA introduces the **forgetting mechanism** (implemented by spatial-temporal token dropping) that decouples computation from the input sequence length, so training and inference costs no longer rise linearly with video length. **MemAug module** (implemented by a cross-attention mechanism) is designed to dynamically recover lost video tokens from the **memory repository**. Without the MemAug module, purely compression‑based approaches inevitably incur irreversible context loss, which leads to sub-optimal performance. As a result, by coupling the forgetting mechanism with MemAug, Hour‑LLaVA delivers compression‑level efficiency while still preserving the full video context sought by extension methods.
>
> - **Hour‑LLaVA is more than a cross-attention module**: The novelty of Hour-LLaVA lies in the *interaction* of the two tightly coupled components: (1) the forgetting mechanism guarantees training efficiency, and (2) MemAug adaptively re-injects video tokens from the memory repository to recover information loss caused by the forgetting mechanism. However, MemAug (implemented by cross attention) is necessary but not sufficient. Without the forgetting mechanism, the trade‑off between contextual fidelity and training efficiency collapses.
>
> - **Hour‑LLaVA does more than frames/tokens dropping**: its learnable **MemAug** module *re‑injects* informative cues that token dropping would otherwise ignore. Moreover, the ablation study below shows that the MemAug module consistently improves the performance across different token dropping strategies, which highlights that MemAug effectively recovers information lost during compression and enhances the overall video-language modeling.
>
>   | Token Dropping Strategy | MemAug | LongVideoBench  | LVBench         |
>   | ----------------------- | ------ | --------------- | --------------- |
>   | Uniform                 | w/     | 54.0 (**+1.9**) | 40.6 (**+2.3**) |
>   |                         | w/o    | 52.1            | 38.3            |
>   | Random                  | w/     | 52.2 (**+0.5**) | 38.8 (**+0.7**) |
>   |                         | w/o    | 51.7            | 38.1            |
>   | Keyframe                | w/     | 53.5 (**+1.5**) | 40.3 (**+1.4**) |
>   |                         | w/o    | 52.0            | 38.9            |
>   | Question-Guided         | w/     | 53.4 (**+2.4**) | 39.7 (**+2.2**) |
>   |                         | w/o    | 51.0            | 37.5            |
>
> **W2.2**: Lack of comparison with the latest related work (Vamba).
>
> **A2.2:** Thanks for pointing out the latest related work, Vamba []. We will cite this work and include a direct comparison in the revision. The comparison with Vamba shows that both Hour‑LLaVA variants (3B and 7B) consistently outperform the considerably larger Vamba‑10B across all three long‑video benchmarks.
>
> | Method        | LVBench | LongVideoBench | VideoMME (w/o subtitles) |
> | ------------- | ------- | -------------- | ------------------------ |
> | Hour-LLaVA-3B | 44.7    | 57.8           | 60.6                     |
> | Hour-LLaVA-7B | **45.6**   | **60.4**           | **63.6**                     |
> | Vamba-10B     | 42.1    | 55.9           | 57.8                     |
>
> **Reference**:
>
> [1] HallusionBench: An Advanced Diagnostic Suite for Entangled Language Hallucination and Visual Illusion in Large Vision-Language Models. CVPR 2024.
>
> [2] VAMBA: Understanding Hour-Long Videos with Hybrid Mamba-Transformers. ICCV 2025.

---

> > ### Author Response · Authors · 2025-08-03
> > **Looking Forward to Your Feedback**
> >
> > Dear Reviewer 6t1N,
> >
> > It seems we have not received your feedback on our response yet. As the discussion period is nearing its halfway point, we wish to ensure that we have addressed all of your concerns as thoroughly as possible.
> >
> > If there are any remaining questions or points requiring further clarification, we would be most grateful for the opportunity to address them.
> >
> > We sincerely look forward to your feedback. Thanks again for your time!

---

> > > ### Comment · Area_Chair_XrZk · 2025-08-06
> > > **Re: Rebuttal by Authors**
> > >
> > > Dear Reviewer 6t1N,
> > >
> > > could you please let us know your feedback with respect to the authors rebuttal?
> > >
> > > Best!
> > > Your friendly AC

---

> > > > ### Author Response · Authors · 2025-08-07
> > > > **Follow Up: Looking Forward to Your Feedback**
> > > >
> > > > Dear Reviewer 6t1N,
> > > >
> > > > It seems we have not received your feedback on our response yet. As the discussion period will end soon, we were wondering if there is anything else we could address to help further clarify? We sincerely look forward to your feedback. Thanks again for your time!

---

> > > > > ### Author Response · Authors · 2025-08-09
> > > > > **Follow Up: Looking Forward to Your Feedback**
> > > > >
> > > > > Dear Reviewer 6t1N,
> > > > >
> > > > > It seems we have not received your feedback on our response yet. As the discussion period will end soon, we were wondering if there is anything else we could address to help further clarify? We sincerely look forward to your feedback. Thanks again for your time!

---

### Author Response · Authors · 2025-08-06
**Updates on Hallucination Evaluation for Open-Ended QA Samples**

Dear Reviewers 6t1N, K5fQ, and 6Aid,

Thank you for raising concerns regarding the need for human evaluation. Beyond evaluating the **accuracy** in Multiple-Choice QA samples and **Yes/No Bias** on Open-Ended QA samples, we further update the hallucination evaluation setting for **Open-Ended QA Samples** following the suggestions by Reviewer **6Aid**.

Specifically, we refine the hallucination evaluation setting for Open-Ended (OE) QA samples by adopting a more fine-grained 1–10 scale. Here, Score 1 denotes a totally irrelevant response, while Score 10 indicates an almost perfect answer with no hallucinations. Higher scores correspond to fewer hallucinations.

Using the more fine-grained scale, human evaluation achieves an average score of **7.81 ± 0.59** (based on ratings from 20 volunteers over 50 OE QA samples), validating the quality of OE QA samples in the VideoMarathon dataset (Scores above **7.5** are considered *largely correct*).

Furthermore, we report the score distribution to further validate reliability. Notably, over **80%** of the samples receive scores ≥ 7.

| Score         | 1    | 2    | 3    | 4    | 5    | 6    | 7     | 8     | 9     | 10    |
| ------------- | ---- | ---- | ---- | ---- | ---- | ---- | ----- | ----- | ----- | ----- |
| **Ratio **(%) | 2.22 | 2.44 | 1.78 | 3.56 | 4.67 | 4.67 | 13.33 | 22.67 | 18.00 | 26.67 |

To further support the claim "**Scores above 7.5 are considered *largely correct***", we present examples with averaged scores between 7–8:

```
# Example 1
Question: Where is the woman positioned relative to the man in the scene where they are sitting on the couch?
Synthetic Answer: The woman is sitting next to the man on the couch.
Ground Truth: The woman is sitting to the right of the man on the couch.
Averaged Score: 8.0

# Example 2
Question: After the boy lights the candle in Clip, what does the woman do?
Synthetic Answer: She holds the candle after it is lit.
Ground Truth: She starts to speak and holds the candle.
Averaged Score: 7.3

# Example 3
Question: What is the woman doing when she uses a sewing machine in the video?
Synthetic Answer: She is sewing the cuffed hem onto the pants.
Ground Truth: She is sewing the hem of the pants.
Averaged Score: 7.5
```

---

### Author Response · Authors · 2025-08-09
**General Response (2/2)**

We have carefully addressed each of the reviewers’ comments. The **key revisions** are summarized below:

- **[Novelty]** (from Reviewers **6t1N** and **K5fQ**) We have provided a thorough discussion on the **novelty** of our proposed approach in response to comments from Reviewer **6t1N** and **K5fQ**. In brief, the novelty of this work lies in two perspectives: (1) **Dataset**: VideoMarathon advances long-video instruction data and bridges the gap between training and inference video lengths with longer averages, broader durations, and more diverse QA pairs than prior training datasets; and (2) **Methodology**: Hour-LLaVA balances training efficiency and video context fidelity to learn effectively from hour-scale videos by the interaction between MemAug module and forgetting mechanism.
- **[Human Evaluation]** (from Reviewers **6t1N**, **K5fQ**, and **6Aid**) We have conducted a comprehensive **human evaluation** of VideoMarathon by manually reviewing 330 randomly sampled QA pairs (80 multiple-choice, 250 open-ended). For *multiple-choice QA* samples, we measured **accuracy**. For *open-ended QA* samples, we have assessed **yes/no bias** and have organized 20 volunteers to rate **hallucination** on a 1–10 scale.
- **[Dataset]** (from Reviewer **K5fQ**) We have clarified that the 22-task taxonomy is a distilled summary of those tasks across a wide range of benchmarks, aiming to provide a **comprehensive and structured categorization** of video-language understanding tasks.
- **[Dataset]** (from Reviewer **K5fQ**) We have elaborated on the **significance of synthetic data** in the video-language domain. Better LMMs produce higher-quality synthetic data, which in turn further improves subsequent LMMs.
- **[Methodology]** (from Reviewer **K5fQ**) We have elaborated the factors that benefit the **generalization** of Hour-LLaVA for **videos longer than the training period**. Moreover, we have presented empirical results showing that Hour-LLaVA exhibits stronger generalization than both its pretrained model and the peer model trained on short videos.
- **[Methodology]** (from Reviewer **K5fQ**) We have clarified that **the purpose of the forgetting mechanism** is to compress the amount of visual tokens into an affordable number, enabling the LLM decoder to operate directly on fewer tokens.
- **[Methodology]** (from Reviewers **ozeC**) We have elaborated on how we use the structure RoPE to bridge the **spatial-temporal relationship** between the *full* video tokens in the memory repository and *decayed* video tokens.
- **[Experiment]** (from Reviewers **6Aid**) We have evaluated Hour-LLaVA-3B and Hour-LLaVA-7B on MLVU-Dev [1] with **free-form generation tasks**.
- **[Experiment]** (from Reviewers **6Aid**) We have conducted an **ablation study on the number of MemAug blocks**.
- **[Experiment]** (from Reviewers **6Aid**) We have added the **comparisons to advanced Video-LMMs**, including Qwen2-VL, Qwen2.5-VL, InternVL3, InternVideo2.5, and LLaVA-OV-SI.
- **[Experiment]** (from Reviewer **K5fQ**) We have compared the **memory usage** of Hour-LLaVA with its baseline when processing different numbers of frames.
- **[Experiment]** (from Reviewer **K5fQ**) We have presented a **complexity comparison** between Hour-LLaVA and its base model.
- **[Experiment]** (from Reviewer **K5fQ**) We have provided an **empirical** analysis to identify a **suitable compression range**.
- **[Experiment]** (from Reviewers **6t1N**) We have added the comparison with the latest related work, **Vamba**.
- **[Experiment]** (from Reviewers **ozeC**) We have added results on **temporal reasoning** tasks.

We are grateful for the thoughtful and constructive feedback, which has significantly improved the quality of our work. We hope these revisions have addressed all concerns. Since Reviewer 6t1N has not yet responded, we would be happy to provide further clarification if needed.

Thank you,

The Authors

---

### Author Response · Authors · 2025-08-09
**General Response (1/2)**

Dear Reviewers (6t1N, K5fQ, 6Aid, ozeC), AC, SAC, and PC,

Thank you for your thoughtful reviews and insightful suggestions. We appreciate that you recognized the following strengths of our work:

1. **Significance of VideoMarathon dataset**: The proposed training dataset advances long-video instruction data and bridges the gap between training and inference video lengths. The hierarchical captioning pipeline and task-specific QA generation help address data scarcity for hour-scale modeling. (Reviewers **6t1N, K5fQ, 6Aid, ozeC**)
2. **Dataset Quality**: The hierarchical captioning pipeline (clip → event → global) is well-designed and leverages appropriate models (Qwen2VL-7B for captioning, DeepSeek-V3 for summarization). (Reviewers **K5fQ**)
3. **Novelty and Superiority of Methodology**: The proposed memory repository, forgetting mechanism, and MemAug module seem novel. As lots of works try to compress the video tokens, the integration of a memory repository and cross-attention is needed, which behaves like retrieval to fetch detailed information if needed. (Reviewers **ozeC**). The MemAug mechanism is technically sound. Ablations validate MemAug’s superiority over heuristic compression methods. (Reviewers **K5fQ, 6Aid**).
4. **Strong Empirical Results** – Architecture achieves good results. Experiments are comprehensive, covering multiple benchmarks with diverse video lengths. (Reviewers **6t1N, K5fQ, ozeC**).
5. **Clarity** – The paper is well-written with clear motivation and methodology. Figures effectively illustrate the architecture and dataset pipeline. (Reviewers **K5fQ, ozeC**).

---

### Decision · Program_Chairs · 2025-09-17

**Decision:**

Accept (spotlight)

**Comment:**

The paper deals with the problem of hour-long Video-LLMs and proposes a new dataset for instruction fine-tuning, VideoMarathon (3.3M  QA pairs. To address the challenge, the paper features a new framework, Hour-LLaVA, based on memory augmentation, which achieves SOTA performance on multiple long video-language benchmarks.

The paper was considered by four reviewers. The final ratings were: BA - BA - BA - A

All reviewers highlight the contribution of the dataset as well as of the method.
The AC follows the consensus of the reviewer voting and recommends accepting the paper. The AC would encourage the authors to integrate the findings of the rebuttal in the CR version of the paper.